# Random Matrix Analysis to Balance between Supervised and Unsupervised Learning under the Low Density Separation Assumption

## Abstract

We propose a theoretical framework to analyze semi-supervised classification under the low density separation assumption in a high-dimensional regime. In particular, we introduce QLDS, a linear classification model, where the low density separation assumption is implemented via quadratic margin maximization. The algorithm has an explicit solution with rich theoretical properties, and we show that particular cases of our algorithm are the least-square support vector machine in the supervised case, the spectral clustering in the fully unsupervised regime, and a class of semi-supervised graph-based approaches. As such, QLDS establishes a smooth bridge between these supervised and unsupervised learning methods. Using recent advances in the random matrix theory, we formally derive a theoretical evaluation of the classification error in the asymptotic regime. As an application, we derive a hyperparameter selection policy that finds the best balance between the supervised and the unsupervised terms of our learning criterion. Finally, we provide extensive illustrations of our framework, as well as an experimental study on several benchmarks to demonstrate that QLDS, while being computationally more efficient, improves over cross-validation for hyperparameter selection, indicating a high promise of the usage of random matrix theory for semi-supervised model selection.

## 1 Introduction

Semi-supervised learning (SSL, Chapelle et al., 2010; van Engelen and Hoos, 2019) aims to learn using both labeled and unlabeled data at once. This machine learning approach received a lot of attention over the past decade due to its relevance to many real-world applications, where the annotation of data is costly and performed manually (Imran et al., 2020), while the data acquisition is cheap and may result in an abundance of unlabeled data (Fergus et al., 2009). As such, semi-supervised learning could be seen as a learning framework that lies in between the supervised and the unsupervised settings, where the former occurs when all the data is labeled, and the latter is restored when only unlabeled data is available. Generally, a semi-supervised algorithm is expected to outperform its supervised counterpart trained only on labeled data by efficiently extracting the information valuable to the prediction task from unlabeled examples.

In practice, integration of unlabeled observations to the learning process does not always affect the performance (Singh et al., 2008), since the marginal data distribution $p(\mathbf{x})$ must contain information on the prediction task $p(y|\mathbf{x})$. Consequently, most semi-supervised approaches rely on specific assumptions about how $p(\mathbf{x})$ and $p(y|\mathbf{x})$ are linked with each other. It is principally assumed that examples *similar* to each other tend to share the same class labels (van Engelen and Hoos, 2019), and implementation of this assumption results in different families of semi-supervised learning models. The first approaches aim to capture the intrinsic geometry of the data using a graph Laplacian (Chong et al., 2020; Song et al., 2022) and suppose that high-dimensional data points with the same label lie on the same low-dimensional *manifold* (Belkin and Niyogi, 2004). Another family of semi-supervised algorithms suggests that examples from a dense region belong to the same class. While some methods explicitly look for such regions by relying on a clustering algorithm (Rigollet, 2007; Peikari et al., 2018), another idea is to directly restrict the classification model to have a decision boundary that only passes through low density regions. This latter approach is said to rely on the *Low Density*

*Separation* (LDS) assumption (Chapelle and Zien, 2005; van Engelen and Hoos, 2019), and it has been widely used in practice in recent decades, combined with the support vector machine (Bennett and Demiriz, 1998; Joachims, 1999), ensemble methods (d'Alché-Buc et al., 2001; Feofanov et al., 2019) and deep learning methods (Sajjadi et al., 2016; Berthelot et al., 2019).

Despite its popularity, the study of the low density separation assumption still has many open questions. First, there is a deficiency of works devoted to theoretical analysis of the algorithm's performance under this assumption, and most approaches focus on the methodological part (van Engelen and Hoos, 2019). Second, in real applications, it always remains unclear how a semi-supervised algorithm should balance the importance of the labeled and the unlabeled examples in order to not degrade the performance with respect to supervised and unsupervised baselines. This implies in particular that the hyperparameter selection for a semi-supervised classification model is crucial, and it is known that using the cross-validation for model selection may be suboptimal in the semi-supervised case due to the lack of labeled examples (Madani et al., 2005).

Motivated by the aforementioned reasons, this paper proposes a framework to analyze semi-supervised classification under the low density separation assumption using the power of the random matrix theory (Paul and Aue, 2014; Marchenko and Pastur, 1967). We consider a simple yet insightful quadratic margin maximization problem, QLDS, that seeks for an optimal balance between the labeled part represented by the Least Square Support Vector Machine (LS-SVM, Suykens and Vandewalle, 1999) and the unlabeled part represented by the spectral clustering (Ng et al., 2001). In addition, the considered algorithm recovers the graph-based approach proposed by Mai and Couillet (2021) as a particular case.

The main contributions of this paper may be summarized as follows:

- We propose a large dimensional analysis of QLDS and derive a theoretical evaluation of the classification error in the asymptotic regime under the data concentration assumption (Louart and Couillet, 2018). The results allow a strong understanding of the interplay between the data statistics and the hyperparameters of the model.

- Based on the proposed theoretical result, we propose a hyperparameter selection approach to optimally balance the supervised and unsupervised term of QLDS. We empirically validate this approach on synthetic and real-world data showing that it outperforms a hyperparameter selection by the cross-validation both in terms of performance and running time.

The remainder of the article is structured as follows. In Section 2, we review the related work. Section 3 introduces the semi-supervised framework as well as the optimization problem of QLDS. Under mild conditions on the data distribution, Section 4 provides the large dimensional analysis of the proposed algorithm along with several insights and discusses its application for hyperparameter selection. Section 5 provides various numerical experiments to corroborate the pertinence of the theoretical analysis and to hyperparameter selection policy. Section 6 concludes the article.

## 2  RELATED WORK

**LDS in Semi-supervised Learning.** Formally introduced by Chapelle and Zien (2005), the LDS assumption imposes the optimal class boundary to lie in a low density region. This assumption is usually implemented by margin maximization, which underlies either explicitly or implicitly many semi-supervised algorithms such as the Transductive SVM (TSVM) (Joachims, 1999; Ding et al., 2017), self-training (Tür et al., 2005; Feofanov et al., 2019) or entropy minimization approaches (Grandvalet and Bengio, 2004; Sajjadi et al., 2016). As the margin's signs for unlabeled data are unknown, various unsigned alternatives have been proposed (d'Alché-Buc et al., 2001; Grandvalet and Bengio, 2004), where the classical approach is to consider the margin's absolute value (Joachims, 1999; Amini et al., 2008). In practice, the latter is usually replaced by an exponential surrogate function for gradient-based optimization of TSVM (Chapelle and Zien, 2005; Gieseke et al., 2014). In this paper, we will consider TSVM with the quadratic margin that is both differentiable and convex, which allows us to perform theoretical analysis and obtain a graph-based semi-supervised learning as a particular case. A similar framework of the quadratic margins was considered by Belkin et al. (2006) whose work considered a more general case with a kernel-based SVM and the Laplacian matrix integrated to the objective, for which they proved a Representer theorem. While our work focuses on explicitly deriving a theoretical expression of the classification error, their paper may complement

us from the algorithmic point of view showing a direct extension of QLDS to a non-linear case. It is important to mention other theoretical studies of approaches based on the low density separation, including upper-bounds of the classification error of TSVM (Derbeko et al., 2004; Wang et al., 2007) and analysis of the self-training algorithm (Feofanov et al., 2021; Zhang et al., 2022).

**Graph-based Semi-Supervised Learning.** The principle of a graph-based approach is to 1) build a suitable graph with all the labeled and the unlabeled examples as the nodes connected by the weighted edges measuring the pairwise similarities (graph construction step), 2) search for a function $f$ over the graph that is close as possible to the given labels, and that is smooth on the entire constructed graph (label inference step). The graph structure can be naturally used as a reflection for the manifold assumption in SSL that suggests that samples located near to each other on a low-dimensional manifold should share similar labels. Among the graph construction methods, the K-nearest neighbor (KNN) graph (Ozaki et al., 2011; Vega-Oliveros et al., 2014) and b-Matching methods (Jebara et al., 2009; Dhillon et al., 2010), along with their extensions, are the most popular ones. Several extensions have considered labeled samples as prior knowledge to refine the generated graph (Rohban and Rabiee, 2012; Berton and Lopes, 2014). Depending on the particular choice of loss functions, the label inference methods can be divided in label propagation approaches (Xiaojin and Zoubin, 2002; Zhou et al., 2003), manifold regularization (Belkin et al., 2006; Xu et al., 2010), Poisson learning (Calder et al., 2020) and deformed Laplacian regularization (Gong et al., 2015). Recently, Mai and Couillet (2021) proposed a theoretical analysis of a unified framework for label inference in a graph that encompasses label propagation, manifold, and Laplacian regularization as special cases. In this paper, we recover (Mai and Couillet, 2021) as a special case of QLDS.

**Large Dimensional Analysis for Machine Learning.** Recently, Random Matrix Theory (RMT) has received particular attention in the machine learning community for studying the asymptotic performance in a regime when the dimension is of the same order of magnitude as the sample size. Recent advances include analysis of the linear discriminant (Niyazi et al., 2021), spectral clustering (Couillet and Benaych-Georges, 2016), least square SVM (Liao and Couillet, 2019), graph-based semi-supervised learning (Mai and Couillet, 2021). In this paper, we show that theoretical findings of the last three aforementioned works are recovered from our theoretical analysis of QLDS provided in Section 4. We derive our theoretical results under the assumption that observations follow a vector-concentration inequality (Louart and Couillet, 2018), which can be particularly interesting for deep learning representations that preserve concentration property (Seddik et al., 2020). It is interesting to mention that a number of machine learning algorithms have been theoretically analyzed using methods from theoretical physics, especially glassy physics (Agliari et al., 2020; Carleo et al., 2019; Loureiro et al., 2021; Cui et al., 2021; d'Ascoli et al., 2020). To continue with physical statistics-based methods, we highlight the work of Lelarge and Miolane (2019) who derived Asymptotic Bayes risk using information theory and the cavity method (Mézard et al., 1987). Although statistical physics and RMT-based approaches share the same objectives, the techniques used and the interpretations make them two different but complementary methods. To the best of our knowledge, we are not aware of any analysis of the algorithm studied in this paper using a statistical physics approach, which we believe is however possible and could be an interesting future work. For completeness, let us also mention the works based on the Convex Gaussian MinMax Theorem (Thrampoulidis et al., 2015; 2016) that allows the analysis of many machine learning algorithms but is mathematically different from the approach used in this paper.

## 3 FRAMEWORK

**Notations** Matrices will be represented by bold capital letters (*e.g.,* matrix $\mathbf{A}$). Vectors will be represented in bold minuscule letters (*e.g.,* vector $\mathbf{v}$) and scalars will be represented without bold letters (*e.g.,* variable $a$). The *canonical vector* of size $n$ is denoted by $\mathbf{e}_m^{[n]} \in \mathbb{R}^n$, $1 \leq m \leq n$, where the $i$-th element is 1 if $i = m$, and 0 otherwise. The diagonal matrix with diagonal $\mathbf{x}$ and 0 elsewhere is denoted by $\mathcal{D}_{\mathbf{x}}$, while $A_{i:}$ denotes the $i$-th line of the matrix $A$.

**Semi-supervised Setting** We consider binary classification problems, where an observation $\mathbf{x} \in \mathbb{R}^d$ is described by $d$ features and belongs either to the class $\mathcal{C}_1$ with a label $y = -1$ or to the class $\mathcal{C}_2$ with a label $y = +1$. We assume that training data consists of $n_l$ labeled examples $(\mathbf{X}_\ell, \mathbf{y}_\ell) =$

$(\mathbf{x}_i, y_i)_{i=1}^{n_\ell} \in \mathbb{R}^{d \times n_\ell} \times \{-1, +1\}^{n_\ell}$ and $n_u$ unlabeled examples $\mathbf{X}_u = (\mathbf{x}_i)_{i=n_\ell+1}^{n_\ell+n_u} \in \mathbb{R}^{d \times n_u}$ given without labels. Following the transductive setting (Vapnik, 1982), we formulate the goal of semi-supervised learning as to learn a classification model $\mathbb{R}^d \to \{-1, +1\}$ that yields the minimal error on the unlabeled data $\mathbf{X}_u$. For convenience, we denote the concatenation of labeled and unlabeled observations by $\mathbf{X} = [\mathbf{X}_\ell, \mathbf{X}_u]$. For each class $\mathcal{C}_j$, $j \in \{1, 2\}$, we denote the observations from this class as $\mathbf{X}_\ell^{(j)} = [\mathbf{x}_1^{(j)}, \dots, \mathbf{x}_{n_{\ell j}}^{(j)}]$, where $\mathbf{X}_\ell = [\mathbf{X}_\ell^{(1)}, \mathbf{X}_\ell^{(2)}]$ and $n_{\ell 1} + n_{\ell 2} = n_\ell$. The same convention is used for the unlabeled data $\mathbf{X}_u$. By $n_j = n_{\ell j} + n_{uj}$ we denote the total number of samples in class $\mathcal{C}_j$, $j \in \{1, 2\}$.

**QLDS** Based on the training set $[\mathbf{X}_\ell, \mathbf{X}_u]$, we seek for a separating hyperplane (linear decision boundary) $\boldsymbol{\omega}^\star$ that is a solution of the following optimization problem:

$$\boldsymbol{\omega}^\star = \arg\min_{\boldsymbol{\omega}} \underbrace{\frac{\alpha_\ell}{2} \sum_{i=1}^{n_\ell} \left( y_i - \frac{\mathbf{x}_i^\top \boldsymbol{\omega}}{\sqrt{n}} \right)^2}_{\text{label fidelity term}} - \underbrace{\frac{\alpha_u}{2} \sum_{i=n_\ell+1}^{n_\ell+n_u} \left( \boldsymbol{\omega}^\top \frac{\mathbf{x}_i}{\sqrt{n}} \right)^2}_{\text{low density separation}} + \underbrace{\frac{\lambda}{2} \|\boldsymbol{\omega}\|^2}_{\text{regularization}} . \tag{1}$$

The first term is the label fidelity term that involves the labeled data only, and it represents the classical least-square loss used in the LS-SVM. The second term implements the low density separation regularization by maximizing the square of the margin of each unlabeled example, thereby pushing the decision boundary away from the unlabeled points. The third term is the classical Tikhonov regularization although we do fix $\lambda$ to the maximum eigenvalue of $\mathbf{X} = [\mathbf{X}_\ell, \mathbf{X}_u]$ (for more details, see Appendix C.2 and E.6). The first two terms are considered up to a $(1/\sqrt{n})$ factor in order to ease the notations of the theoretical derivations of Section 4.

Note that the label fidelity term can be alternatively represented by the hinge loss or the log-loss, which slightly alters the overall behavior of the algorithm. Our choice of the least square loss is primarily motivated by the possibility of obtaining more explicit, tractable and insightful results, let alone numerically cheaper implementation. The question of the optimal choice for the loss of the supervised part is a highly interesting question in the literature. Although it is difficult to formulate a strong statement valid for all practical situations, some asymptotic attempts have been made such as (Aubin et al., 2020; Mai and Liao, 2019). More related to our hypothesis, (Mai and Liao, 2019) shows that for isotropic Gaussian mixture models in the high dimensional regime, quadratic cost functions are optimal and outperform alternatives costs such as SVM or logistic approaches. Table 4 in Appendix summarizes the classification error by using three losses for labelled parts (hinge, logistic, and quadratic) and two losses for unlabelled parts (quadratic and absolute value). This table shows that the selection of losses presented in the article has a competitive performance.

The optimization problem in Equation (1) is convex (as soon as $\lambda > \lambda_{max}$ where $\lambda_{max}$ is the maximum eigenvalue of $\left( \alpha_u \frac{\mathbf{X}_u \mathbf{X}_u^\top}{n} - \alpha_\ell \frac{\mathbf{X}_\ell \mathbf{X}_\ell^\top}{n} \right)$) and admits a unique solution (all details are given in the supplementary material, Section A) given by

$$\boldsymbol{\omega}^\star = \frac{1}{\sqrt{n}} \left( \lambda \mathbf{I}_d - \alpha_u \frac{\mathbf{X}_u \mathbf{X}_u^\top}{n} + \alpha_\ell \frac{\mathbf{X}_\ell \mathbf{X}_\ell^\top}{n} \right)^{-1} \mathbf{X}_\ell \mathbf{y}_\ell . \tag{2}$$

It is worth remarking that for the fully-supervised case $(\alpha_\ell, \alpha_u) = (1, 0)$, we recover the Least Square SVM (Suykens and Vandewalle, 1999). Another extreme case is to take $(\alpha_\ell, \alpha_u) = (0, 1)$ that leads to the optimal decision boundary of the graph-based approach proposed by Mai and Couillet (2021) (further denoted by *GB-SSL*). Moreover, if additionally to $(\alpha_\ell, \alpha_u) = (0, 1)$ take $\lambda$ as the maximum eigenvalue of the unlabeled data $(1/n) \mathbf{X}_u \mathbf{X}_u^\top$, we recover spectral clustering (See Section B of the supplementary material for a complete derivation).

Given the optimal decision boundary as per Equation (2), the decision score function for any example $\mathbf{x} \in \mathbb{R}^d$ is given as

$$f(\mathbf{x}) = \frac{1}{\sqrt{n}} \boldsymbol{\omega}^{\star\top} \mathbf{x} = \frac{1}{n} \mathbf{y}_\ell^\top \mathbf{X}_\ell^\top \left( \lambda \mathbf{I}_d - \alpha_u \frac{\mathbf{X}_u \mathbf{X}_u^\top}{n} + \alpha_\ell \frac{\mathbf{X}_\ell \mathbf{X}_\ell^\top}{n} \right)^{-1} \mathbf{x} . \tag{3}$$

# 4 THEORETICAL ANALYSIS AND ITS APPLICATION

In this section, we theoretically analyze the statistical behavior of QLDS and its decision function $f(\mathbf{x})$. First, we state the assumptions used for theoretical analysis. Then, we present the main results and describe an application for hyperparameter selection.

## 4.1 ASSUMPTIONS

In the following, we assume the following classical concentration property.

**Assumption 1 (Concentration of $\mathcal{D}(\mathbf{X})$)** *For two classes $\mathcal{C}_j$, $j \in \{1, 2\}$, we assume that all vectors $\mathbf{x}_1^{(j)}, \ldots, \mathbf{x}_{n_j}^{(j)} \in \mathcal{C}_j$ are i.i.d. and in particular $\mathrm{Cov}(\mathbf{x}_i^{(j)}) = \mathbf{I}_d$. Moreover we assume that there exist two constants $C, c > 0$ (independent of $n, d$) such that, for any 1-Lipschitz function $f : \mathbb{R}^d \to \mathbb{R}$,*

$$\forall t > 0, \qquad \mathbb{P}_{\mathbf{x} \sim \mathcal{D}(\mathbf{X})} \left( |f(\mathbf{x}) - m_{f(\mathbf{x})}| \geq t \right) \leq C e^{-(t/c)^2}$$

*where $m_Z$ is a median of the random variable $Z$.*

Assumption 1 notably encompasses the following scenarios: the columns of $\mathbf{X}$ are (a) independent Gaussian random vectors with identity covariance, (b) independent random vectors uniformly distributed on the $\mathbb{R}^d$ sphere of radius $\sqrt{d}$, and, most importantly, (c) any Lipschitz continuous transformation thereof, such as GAN as it has been recently theoretically shown in (Seddik et al., 2020). In the appendix (Section D), we have further explained the concentrated vector assumption and complemented its relevance and generality for the study of machine learning algorithms. In Assumption 1, we only consider identity covariance matrix to keep this presentation simple. The more general case of arbitrary covariance matrix $\mathbf{\Sigma}_j$ is fully derived in the supplementary material, Section C. We should mention that it is convenient to "center" the data $\mathbf{X}$ for the sake of simplicity. This centering operation is performed on the whole data set $\mathbf{X}$ by substracting the global mean from the training points *i.e.,* $\mathbf{X} \leftarrow \mathbf{X} - \mathbb{E}[\mathbf{X}]$. Furthermore, we place ourselves into the following large dimensional regime:

**Assumption 2 (High-dimensional asymptotics)** *As $n \to \infty$, we consider the regime where $d = \mathcal{O}(n)$ and assume $d/n \to c_0 > 0$. Furthermore, for $j = 1, 2$, $n_{\ell j}/n \to c_{\ell j}$ and $n_{uj}/n \to c_{uj}$. We denote by $\mathbf{c}_\ell = [c_{\ell 1}, c_{\ell 2}]$ and $\mathbf{c}_u = [c_{u1}, c_{u2}]$.*

This assumption of the commensurable relationship between the number of samples and their dimension corresponds to a realistic regime and differs from classical asymptotic where the number of samples is often assumed to be exponentially larger than the feature size. Note that this chosen asymptotic regime classical in Random Matrix Theory fits most real-life applications and has been successfully applied in telecommunications (Couillet and Debbah, 2011), finance (Potters et al., 2005) and more recently in machine learning (Liao, 2019; Mai and Couillet, 2021; Tiomoko et al., 2020).

## 4.2 MAIN RESULTS

We introduce the mean matrix $\mathbf{M} = [\boldsymbol{\mu}_1, \boldsymbol{\mu}_2] \in \mathbb{R}^{d \times 2}$, where $\boldsymbol{\mu}_j = \mathbb{E}_{\mathbf{x} \in \mathcal{C}_j}[\mathbf{x}] \in \mathbb{R}^d$ is the theoretical mean of the class $\mathcal{C}_j, j \in \{1, 2\}$. Further, we define matrices $\mathcal{M}$ and $\mathcal{G}$ that will play an important role at the core formulation of the statistics of $f(\mathbf{x})$.

**Definition 1 (Data statistics matrices $\mathcal{M}$ and $\mathcal{G}$)** *We define data matrices $\mathcal{M}$ and $\mathcal{G}$ as*

$$\mathcal{M} = \left( \mathcal{D}_{\boldsymbol{\kappa}}^{-1} + \delta \mathbf{M}^\top \mathbf{M} \right)^{-1}, \quad \mathcal{G} = - \left( \frac{n_u}{n(1 - \alpha_u \delta)} + \mathbf{a}^\top \mathbf{d} \right) \delta \mathbf{M}^\top \mathbf{M},$$

*where the vectors $\mathbf{a}$, $\mathbf{d}$ and $\boldsymbol{\kappa}$ are the unique positive solution of the following fixed point equations*

$$a_j = \frac{c_{\ell j} \alpha_\ell^2}{(1 + \alpha_\ell \delta)^2} + \frac{c_{uj} \alpha_u^2}{(1 - \alpha_u \delta)^2}, \quad d_j = -\frac{\delta^2}{(1 - \alpha_u \delta)^2} \frac{c_0 n_u}{n(1 - c_0 \delta^2 a_j)},$$

$$\delta = \frac{1}{\lambda + \kappa_1 + \kappa_2}, \quad \kappa_j = \frac{c_{\ell j} \alpha_\ell}{1 + \alpha_\ell \delta} - \frac{c_{uj} \alpha_u}{1 - \alpha_u \delta}.$$

The existence of $\mathbf{a}, \mathbf{d}, \boldsymbol{\kappa}, \delta$ are a direct application of (Louart and Couillet, 2018, Proposition 3.8)). These quantities are common in Random Matrix Theory in order to correct large biases in high dimensions (for more details, we refer to the supplementary material, Section D). We are now in position to introduce the asymptotic theoretical analysis of the score $f(\mathbf{x})$ of any unlabeled sample $\mathbf{x}$.

**Theorem 1** *Let* $\mathbf{X} \in \mathbb{R}^{d \times n}$ *be a data set that follows Assumptions 1 and 2. For any* $\mathbf{x} \in \mathbf{X}_u$ *with* $\mathbf{x} \in \mathcal{C}_j$ *and* $f(\mathbf{x}) = \frac{1}{\sqrt{n}} \boldsymbol{\omega}^{\star \top} \mathbf{x}$ *defined by Equation* (3)*, we have almost surely for both classes* $j$

$$f(\mathbf{x}|\mathbf{x} \in \mathcal{C}_j) - \mathfrak{f}_j \xrightarrow{\text{a.s.}} 0, \quad \text{where} \quad \mathfrak{f}_j \sim \mathcal{N}\left(m_j, \sigma^2\right).$$

*The mean* $m_j$ *and the variance* $\sigma^2$ *are defined as*

$$m_j = \frac{(-1)^j \left(c_{\ell j} - (\mathbf{e}_1^{[2]} - \mathbf{e}_2^{[2]})^\top \mathcal{D}_{\mathbf{c}_\ell} \mathcal{D}_{\boldsymbol{\kappa}}^{-1} \mathcal{M} \mathbf{e}_j^{[2]}\right)}{\kappa_j (1 - \alpha_u \delta)(1 + \alpha_\ell \delta)},$$

$$\sigma^2 = \left(\mathbf{e}_1^{[2]} - \mathbf{e}_2^{[2]}\right)^\top \left(\mathcal{D}_{\mathbf{s}} \mathcal{M} \mathcal{G} \mathcal{M} \mathcal{D}_{\mathbf{s}} + \mathcal{D}_{\mathbf{d}} \mathcal{D}_{\mathbf{c}_\ell}\right) \left(\mathbf{e}_1^{[2]} - \mathbf{e}_2^{[2]}\right),$$

*with* $\mathbf{s} = [c_{\ell 1}/(\kappa_1(1 + \alpha_\ell \delta)), \ c_{\ell 2}/(\kappa_2(1 + \alpha_\ell \delta))]$.

*Finally, the theoretical classification error is asymptotically given by*

$$\varepsilon_\star = \frac{1}{2}\left(1 - \operatorname{erf}\left(\frac{m_1 - m_2}{2\sqrt{2}\sigma}\right)\right), \tag{4}$$

*where* $\operatorname{erf}(z) = 2/\sqrt{\pi} \int_0^z e^{-t^2} dt$ *is the Gauss error function.*

A fundamental aspect of Theorem 1 is that the performance of the *large dimensional* (large $n$, large $d$) classification problem under consideration merely concentrates into two-dimensional *sufficient statistics*, as all objects defined in the theorem are at most of size 2. All quantities defined in Theorem 1 are a priori known, apart from the proportion of classes in $\mathbf{X}_u$ and the matrix $\mathbf{M}^\top \mathbf{M} \in \mathbb{R}^{2 \times 2}$, whose $(i, j)$-entries are the inner products $\boldsymbol{\mu}_i^\top \boldsymbol{\mu}_j$ that have to be estimated from data. From a practical perspective, these inner products are easily amenable to fast and efficient estimation as per Proposition 2, requiring a few training data samples.

**Proposition 2 (On the estimation of** $m_j$ **and** $\sigma$**)** *The following estimates holds:*

$$\left[\mathbf{M}^\top \mathbf{M}\right]_{ij} = \begin{cases} (4/n_{\ell i}^2) \, \mathbf{1}_{n_{\ell i}}^\top \mathbf{X}_{\ell;1}^{(i)\top} \mathbf{X}_{\ell;2}^{(i)} \mathbf{1}_{n_{\ell i}} + \mathcal{O}\left(1/\sqrt{d \, n_{\ell i}}\right) & \text{if } i = j, \\ (1/\left(n_{\ell i} n_{\ell j}\right)) \, \mathbf{1}_{n_{\ell i}}^\top \mathbf{X}_\ell^{(i)\top} \mathbf{X}_\ell^{(j)} \mathbf{1}_{n_{\ell j}} + \mathcal{O}\left((d \min\{n_{\ell i}, n_{\ell j}\})^{-\frac{1}{2}}\right) & \text{otherwise.} \end{cases}$$

*with* $\mathbf{X}_\ell^{(j)} = [\mathbf{X}_{\ell;1}^{(j)}, \mathbf{X}_{\ell;2}^{(j)}]$ *an even-sized division of* $\mathbf{X}_\ell^{(j)}$.

Note that a single sample (two when $i = j$) per class is sufficient to obtain a consistent estimate for all quantities as long as $d$ is large. In the semi-supervised setting, when only few labeled examples are available, it is thus still possible to estimate $\mathbf{M}^\top \mathbf{M}$. It is important to remark that the convergence rate of the estimation is a quadratic improvement over the convergence rate of the usual central-limit theorem. Finally, to estimate the proportion of classes in the unlabeled set, not known *a priori*, we assume that the distribution of classes to be the same for the labeled and unlabeled data, so that we have $c_{uj} = c_{\ell j} \frac{n_u}{n_\ell}$ for $j \in \{1, 2\}$. We show in the supplementary material (Section E) that this assumption has little impact on the theoretical insights as well as in the experiments.

As an application of Theorem 1, we provide in Figure 5 of the Appendix a "phase diagram" (relative gain with respect to supervised learning as a function of the labeled sample size and the task difficulty) which shows that a non-trivial gain with respect to a fully supervised case is obtained when few labeled samples are available and when the task is difficult. This conclusion is similar to existing conclusion from (Mai and Couillet, 2021; Lelarge and Miolane, 2019).

### 4.3 APPLICATION TO HYPERPARAMETER SELECTION

Following the discussion in Section 3, we obtain that the theorem allows us to recover the asymptotic performance of the spectral clustering, the graph-based approach GB-SSL of Mai and Couillet (2021)

Table 1: Running time comparison between theory-based hyperparameter selector and cross-validation based with 10 folds, with $n_{\ell j} = n_{uj} = d$ for $j \in \{1, 2\}$.

| dimension | 5 | 8 | 16 | 32 | 64 | 128 | 256 | 512 | 1024 | 2048 |
|---|---|---|---|---|---|---|---|---|---|---|
| `QLDS(th)` | .01 s | .01 s | .01 s | .01 s | .017 s | .02 s | .04 s | .17 s | .95 s | 8.58 s |
| `QLDS(cv)` | .32 s | .38 s | .43 s | .64 s | 1.02 s | 2.71 s | 14 s | 95 s | 587 s | 4970 s |

and the LS-SVM (Suykens and Vandewalle, 1999). This generality of the theorem represents an important asset in the unification of some SSL learning schemes. In particular, as the theoretical error can be regarded as a function of $\alpha_\ell$ and $\alpha_u$, below we propose to use Equation (4) as a criterion to automatically select $\alpha_\ell$ and $\alpha_u$ through the grid search over different values. This leads to Algorithm 1. Note that the classification error is invariant to a scaling of $\lambda$ (see Equation (3)). Thus, we fix the value of $\lambda$ in our experiments to be $\lambda_{\max}$ with $\lambda_{\max}$ the maximum eigenvalue of $\mathbf{X} = [\mathbf{X}_\ell, \mathbf{X}_u]$, and optimize only $\alpha_\ell$ and $\alpha_u$. The fixed value corresponds to the one also proposed in (Mai and Couillet, 2021). We give more details about this choice for $\lambda$ and its numerical stability in Appendix C.2.

Our proposition to select $\alpha_\ell$ and $\alpha_u$ by the theorem is motivated by several practical reasons. Firstly, the importance of labeled and unlabeled examples varies, making the graph-based learning more effective in some cases, and the LS-SVM more effective in the others. By properly choosing $\alpha_\ell$ and $\alpha_u$, one can find the best balance between the GB-SSL and LS-SVM. Secondly, the classical approach of selecting hyperparameters by cross-validation suffers from high computational time and prone to bias in the semi-supervised setting due to the scarcity of the labeled set (Madani et al., 2005).

---

**Algorithm 1** QLDS algorithm with optimal selection of $\alpha_\ell$ and $\alpha_u$

---

**Input:** labeled data $\mathbf{X}_\ell$ and unlabeled data $\mathbf{X}_u$

        grid of hyperparameter values $\{(\alpha_\ell^{(t)}, \alpha_u^{(t)})\}_{t=1}^T$

**Preprocessing:** center data $\mathbf{X} \leftarrow \mathbf{X} - \mathbb{E}[\mathbf{X}]$, where $\mathbf{X} = [\mathbf{X}_\ell, \mathbf{X}_u]$

**Output:** estimated label $\hat{y} \in \{-1, 1\}$ for each unlabeled example $\mathbf{x} \in \mathbf{X}_u$

    **estimate** inner product $\mathbf{M}^\top \mathbf{M}$ using Proposition 2

    **choose** $\lambda$ as the maximum eigenvalue of $\frac{1}{n}\mathbf{X}_u \mathbf{X}_u^\top$

    **for** $t = 1, \dots, T$ grid-search steps **do**

       **take** $\alpha_\ell^{(t)}$ and $\alpha_u^{(t)}$

       **estimate** classification error $\varepsilon_\star^{(t)}$ by Theorem 1 with $\alpha_u = \alpha_u^{(t)}$ and $\alpha_\ell = \alpha_\ell^{(t)}$

    **end for**

    **select** $\alpha_u^\star$ and $\alpha_\ell^\star$ by finding $t$ that yields minimal classification error $\varepsilon_\star^{(t)}$

    **compute** the decision score $f(\mathbf{x})$ using Equation (3) with $\alpha_u = \alpha_u^\star$ and $\alpha_\ell = \alpha_\ell^\star$

    **return** label $\hat{y} = \begin{cases} -1 & \text{if } f(\mathbf{x}) < 0 \\ 1 & \text{otherwise} \end{cases}$

---

**Complexity analysis.** Algorithm 1 or `QLDS(th)` may be sequentially described as in 1) training of `QLDS`, 2) estimation of $\mathbf{M}^\top \mathbf{M}$, 3) selection of $\alpha_\ell$ and $\alpha_u$ over the grid. As `QLDS` has an explicit solution, its complexity is equivalent to the computation of the decision scores $f(\mathbf{x})$ which requires solving a system of $n$ linear equations, yielding complexity $\mathcal{O}(n^3)$. The computation of $\mathbf{M}^\top \mathbf{M}$ is of complexity $\mathcal{O}(dn + d)$ (estimation + product). Hyperparameter selection consists of iterating $T$ times the error estimation from Theorem 1 and its complexity is $\mathcal{O}(T)$. Finally, the global complexity of `QLDS(th)` is $\mathcal{O}(n^3)$ in the regime of Assumption 2.

It is important to mention that an alternative way to optimize $\alpha_\ell$ and $\alpha_u$ is cross-validation (`QLDS(cv)`), which requires optimizing `QLDS` for each candidate $(\alpha_\ell^t, \alpha_u^t)$ for $K$ folds, leading to a complexity of $\mathcal{O}(TKn^3)$. This indicates a clear advantage of using Theorem 1 in terms of time complexity as highlighted in Table 1.

## 5    EXPERIMENTAL RESULTS

In this section, we illustrate the robustness of the different algorithms and the optimization of the hyperparameters $\alpha_\ell$ and $\alpha_u$ proposed in the previous section. More specifically, Section 5.1 confirms empirically the robustness of the concentrated random vector assumption on real data by comparing the empirical distribution of the decision score with the theoretical prediction of Theorem 1. Section 5.2 analyzes the performance of QLDS when increasing the number of labeled examples, and Section 5.3 is a benchmark with several baselines for a wide range of real data sets. We perform comparison between the following methods:

- `QLDS(0, 1)` with $\alpha_\ell = 0, \alpha_u = 1$ which stands for the graph-based approach proposed in (Mai and Couillet, 2021);
- `QLDS(1, 0)` with $\alpha_\ell = 1, \alpha_u = 0$ which stands for LS-SVM;
- Self-training with the Least-Square SVM as the base classifier, where the confidence threshold is optimized as proposed by Feofanov et al. (2019) denoted `ST (LS-SVM)`;
- `QLDS(cv)` with model selection of $\alpha_\ell$ and $\alpha_u$ by the 10-fold cross-validation;
- `QLDS(th)` with model selection of $\alpha_\ell$ and $\alpha_u$ performed theoretically using Theorem 1;
- `QLDS(or)` Oracle to measure the efficiency of the proposed algorithm: QLDS, where model selection of $\alpha_\ell$ and $\alpha_u$ is performed on the ground truth (as if the labels for the unlabeled examples would be known). It represents the error-classification lower-bound for the previous approaches.

Through the experimental part, we will use several data sets described as follows (see more details in Section E of the supplementary material):

- `Synthetic:` Gaussian mixture model with $\mathbf{x}_i^{(j)} \sim \mathcal{N}(\boldsymbol{\mu}_j, \mathbf{I}_d)$ with $\boldsymbol{\mu}_1 = -\boldsymbol{\mu}_2$;
- `Amazon Review` data set (McAuley et al., 2015; He and McAuley, 2016) from textual user reviews, positive or negative, on books (`books`), DVDs (`dvd`), electronics (`electronics`), and kitchen (`kitchen`) items respectively. The data is encoded as $d = 400$-dimensional tf-idf feature vectors of bag-of-words unigrams and bigrams;
- `Adult` data set (Kohavi et al., 1996) which consists in predicting whether income exceeds $50\,000$ per year based on census data;
- `Mushrooms` data set from UCI Machine Learning repository (Dua and Graff, 2017) which classifies between poisonous and edible mushrooms based on their physical characteristics;
- `Splice` data set from UCI Machine Learning repository (Dua and Graff, 2017) which aims to recognize two types of splice junctions in DNA sequences.

### 5.1    ROBUSTNESS OF THEORETICAL ANALYSIS TO REAL DATA

This section illustrates the close fit of the theoretical performance (*i.e.,* Theorem 1) on the synthetic and two real-life data sets. To do so, we compare the empirical decision function represented by the histograms on Figure 1 versus the Gaussian statistics $m_j$ and $\sigma^2$ from Theorem 1.

### 5.2    ANALYSIS OF SAMPLE SIZE

Figure 2 represents the classification error as a function of the number of labeled examples. The picture shows that the theoretical model selection outperforms the cross-validation scheme and is close to the oracle selection (which uses the ground truth labels). In general, we observe that `QLDS(th)` is very stable in comparison with the cross-validation selector `QLDS(cv)`.

### 5.3    COMPARATIVE PERFORMANCE ON SEVERAL DATA SETS

This section compares through Table 2 the performance obtained by fixing the number of labeled and unlabeled data on several data sets to analyze the performance of the hyperparameter selection and also to validate the theoretical intuitions formulated in this article.

The experimental results show that

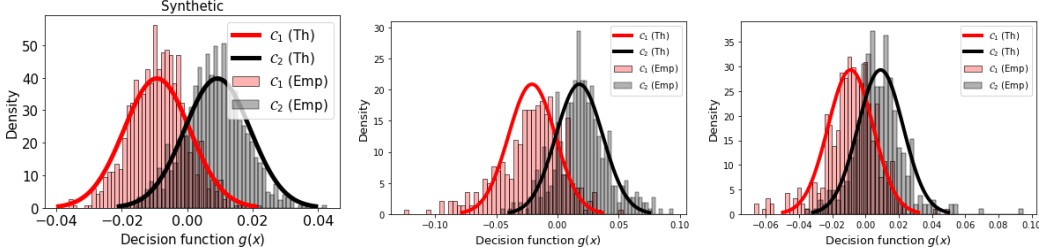

Figure 1: Empirical versus theoretical density of decision score $f(\mathbf{x})$ for (**Left**) Synthetic data set with $d = 100$, $n_{\ell 1} = n_{\ell 2} = 100$, $n_{u1} = n_{u2} = 1\,000$ (**Center**) Review-kitchen classification $d = 400$, $n_{\ell 1} = n_{\ell 2} = 100$ (**Right**) Review-books classification $d = 400$, $n_{\ell 1} = n_{\ell 2} = 100$. For both review data sets, the empirical histogram is computed using $400$ unlabeled samples.

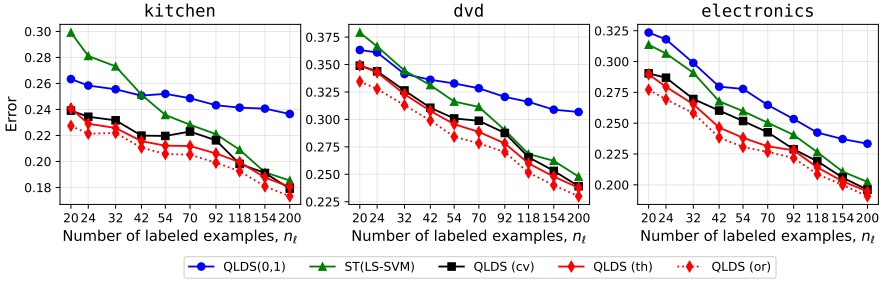

Figure 2: Classification error depending on the number of labeled examples on different data sets. Positive vs. negative review for different products (**Left**) `kitchen` (**Center**) `dvd` and (**Right**) `electronics` with $n_{u1} = n_{u2} = 200$, $d = 400$.

Table 2: The classification error of different methods under consideration on the real benchmark data sets. $^\downarrow$ indicates statistically significantly worse performance than the best result (shown in bold), according to the Mann-Whitney U test ($p < 0.01$) (Mann and Whitney, 1947).

| Data set | Baselines | | | | Model Selection | | Oracle |
|---|---|---|---|---|---|---|---|
| | QLDS (1,0) (LS-SVM) | QLDS (0,1) (GB-SSL) | QLDS (1,1) | ST (LS-SVM) | QLDS (cv) | QLDS (th) | QLDS (or) |
| books | $37.47^\downarrow \pm 2.25$ | $26.47 \pm 0.72$ | $49.13^\downarrow \pm 0.65$ | $35.83^\downarrow \pm 2.48$ | $27.91 \pm 3.32$ | $\mathbf{26.03} \pm 0.79$ | $25.7 \pm 0.93$ |
| dvd | $38.33^\downarrow \pm 1.72$ | $29.12 \pm 1.35$ | $49.25^\downarrow \pm 0.68$ | $36.46^\downarrow \pm 1.94$ | $29.53 \pm 3.48$ | $\mathbf{28.53} \pm 1.33$ | $26.94 \pm 1.47$ |
| electronics | $34.15^\downarrow \pm 3.25$ | $19.4 \pm 0.29$ | $48.67^\downarrow \pm 1.05$ | $31.69^\downarrow \pm 3.56$ | $20.1^\downarrow \pm 1.03$ | $\mathbf{19.41} \pm 0.46$ | $19.11 \pm 0.58$ |
| kitchen | $32.39^\downarrow \pm 3.02$ | $19.31 \pm 0.16$ | $49.07^\downarrow \pm 0.64$ | $29.62^\downarrow \pm 3.03$ | $19.98^\downarrow \pm 2.28$ | $\mathbf{19.11} \pm 0.32$ | $18.67 \pm 0.43$ |
| splice | $39.81^\downarrow \pm 2.93$ | $35.48 \pm 0.86$ | $44.36^\downarrow \pm 2.3$ | $39.36^\downarrow \pm 3.12$ | $37.02 \pm 3.04$ | $\mathbf{35.35} \pm 1.26$ | $33.63 \pm 1.75$ |
| adult | $33.35 \pm 0.68$ | $36.28^\downarrow \pm 0.06$ | $32.55 \pm 1.47$ | $35.45^\downarrow \pm 0.75$ | $\mathbf{32.25} \pm 1.92$ | $32.88 \pm 2.46$ | $31.9 \pm 1.74$ |
| mushrooms | $6.55^\downarrow \pm 2.07$ | $11.33^\downarrow \pm 0.04$ | $33.94^\downarrow \pm 10.67$ | $6.62^\downarrow \pm 2.39$ | $\mathbf{2.57} \pm 1.86$ | $8.49^\downarrow \pm 3.63$ | $1.75 \pm 1.31$ |

- QLDS benefits from both labelled and unlabelled data and significantly outperforms `LS-SVM` and `GB-SSL` on datasets 5 and 2 respectively.

- Fine tunning of $\alpha_\ell$ and $\alpha_u$ provides better results than setting them to the default values.

- Hyperparameter selection using Theorem 1 outperforms or is comparable to cross-validation, at the same time being more robust according to the error's standard deviation.

- There is still room for improvement when we compare `QLDS (or)` with `QLDS (th)`.

## 6 CONCLUDING REMARKS

In this paper, we proposed a theoretical analysis of a simple yet powerful linear semi-supervised classifier that relies on the low density separation assumption. Moreover, our approach builds a bridge

between several existing approaches such as the least square support vector machine, the spectral clustering, and graph-based semi-supervised learning. The key approach to our analysis was to use modern large dimensional statistics to quantify the classification error through the data statistics of the decision function. Based on this result, we proposed a hyperparameter selection criterion that demonstrated promising experimental results compared to the time-consuming cross-validation. The proposed theoretical study opens broad perspectives for analysis of the LDS assumption in more challenging settings such as the multi-class classification, the non-linear case, or fully unsupervised domain adaptation.

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

## A  APPENDIX

## B  SOLUTION OF QLDS

We recall the optimization problem of QLDS *without bias* as

$$\boldsymbol{\omega}^{\star} = \arg\min_{\boldsymbol{\omega}} \mathcal{L}(\boldsymbol{\omega}), \tag{5}$$

$$\text{where} \quad \mathcal{L}(\boldsymbol{\omega}) = \frac{\lambda}{2}\|\boldsymbol{\omega}\|^2 + \frac{\alpha_\ell}{2}\sum_{i=1}^{n_\ell}(y_i - \frac{\mathbf{x}_i^\top}{\sqrt{n}}\boldsymbol{\omega})^2 - \frac{\alpha_u}{2}\sum_{i=n_\ell+1}^{n_\ell+n_u}(\boldsymbol{\omega}^\top\frac{\mathbf{x}_i}{\sqrt{n}})^2. \tag{6}$$

The loss $\mathcal{L}(\boldsymbol{\omega})$ can be rewritten in a more convenient and compact matrix formulation

$$\mathcal{L}(\boldsymbol{\omega}) = \frac{\lambda}{2}\boldsymbol{\omega}^\top\boldsymbol{\omega} + \frac{\alpha_\ell}{2}\|\mathbf{y}_\ell - \frac{\mathbf{X}_\ell^\top}{\sqrt{n}}\boldsymbol{\omega}\|_2^2 - \frac{\alpha_u}{2}\boldsymbol{\omega}^\top\frac{\mathbf{X}_u\mathbf{X}_u^\top}{n}\boldsymbol{\omega}. \tag{7}$$

Taking the derivative of the loss function $\mathcal{L}(\boldsymbol{\omega})$ with respect to $\boldsymbol{\omega}$ leads to

$$\frac{\partial\mathcal{L}(\boldsymbol{\omega})}{\partial\boldsymbol{\omega}} = \lambda\boldsymbol{\omega} - \alpha_\ell\frac{\mathbf{X}_\ell}{\sqrt{n}}\left(\mathbf{y}_\ell - \frac{\mathbf{X}_\ell^\top}{\sqrt{n}}\boldsymbol{\omega}\right) - \alpha_u\frac{\mathbf{X}_u\mathbf{X}_u^\top}{n}\boldsymbol{\omega}$$

$$= \left(\lambda\mathbf{I}_d + \alpha_\ell\frac{\mathbf{X}_\ell\mathbf{X}_\ell^\top}{n} - \alpha_u\frac{\mathbf{X}_u\mathbf{X}_u^\top}{n}\right)\boldsymbol{\omega} - \alpha_\ell\frac{\mathbf{X}_\ell}{\sqrt{n}}\mathbf{y}_\ell.$$

The optimal value of $\omega$ (up to a scaling of $\alpha_\ell$) is found by setting the gradient to zero

$$\boldsymbol{\omega}^{\star} = \left(\lambda\mathbf{I}_d + \alpha_\ell\frac{\mathbf{X}_\ell\mathbf{X}_\ell^\top}{n} - \alpha_u\frac{\mathbf{X}_u\mathbf{X}_u^\top}{n}\right)^{-1}\frac{\mathbf{X}_\ell}{\sqrt{n}}\mathbf{y}_\ell.$$

The decision function for the unlabeled data $\mathbf{X}_u$ is given as

$$\mathbf{f}_u = \boldsymbol{\omega}^{\star\top}\frac{\mathbf{X}_u}{\sqrt{n}} \tag{8}$$

$$= \mathbf{y}_\ell^\top\frac{\mathbf{X}_\ell^\top}{\sqrt{n}}\left(\lambda\mathbf{I}_d + \alpha_\ell\frac{\mathbf{X}_\ell\mathbf{X}_\ell^\top}{n} - \alpha_u\frac{\mathbf{X}_u\mathbf{X}_u^\top}{n}\right)^{-1}\frac{\mathbf{X}_u}{\sqrt{n}}. \tag{9}$$

We would like to mention importantly that the hessian of the loss reads as

$$\nabla\mathcal{L}(\boldsymbol{\omega}) = \left(\lambda\mathbf{I}_d + \alpha_\ell\frac{\mathbf{X}_\ell\mathbf{X}_\ell^\top}{n} - \alpha_u\frac{\mathbf{X}_u\mathbf{X}_u^\top}{n}\right)$$

Note that $\nabla\mathcal{L}(\boldsymbol{\omega}) > 0$ if and only if $\lambda > \lambda_{max}\left(-\alpha_\ell\frac{\mathbf{X}_\ell\mathbf{X}_\ell^\top}{n} + \alpha_u\frac{\mathbf{X}_u\mathbf{X}_u^\top}{n}\right)$ where $\lambda_{max}(M)$ denotes the maximum eigenvalue of the matrix $M$. Therefore the loss function is convex as soon as $\lambda > \lambda_{max}\left(-\alpha_\ell\frac{\mathbf{X}_\ell\mathbf{X}_\ell^\top}{n} + \alpha_u\frac{\mathbf{X}_u\mathbf{X}_u^\top}{n}\right)$.

## C  LINK TO RELATED WORK

### C.1  LINK TO GRAPH-BASED SEMI-SUPERVISED LEARNING

Given generally few labeled examples and comparatively many unlabeled ones, the idea of graph-based SSL is to construct a connected graph that propagates effective labeled information to the unlabeled data. More specifically, the data are represented by a finite weighted graph $\mathcal{G} = (\mathcal{N}, \mathcal{E}, \mathbf{W})$ consisting of a set of nodes $\mathcal{N}$ based on the data samples $\mathbf{X} = [\mathbf{X}_\ell, \mathbf{X}_u]$, a set of edges $\mathcal{E}$ and its associated weight matrix $\mathbf{W} = \{\omega_{ii'}\}_{i,i'=1}^n$ where $\omega_{ii'}$ measures the similarity between data points $\mathbf{x}_i$ and $\mathbf{x}_{i'}$

$$\omega_{ii'} = h\left(\frac{1}{d}\langle\mathbf{x}_i, \mathbf{x}_{i'}\rangle\right),$$

for some non decreasing non negative function $h$ so that similar data vectors $\mathbf{x}_i$, $\mathbf{x}_{i'}$ are connected with a large weight. Graph-based learning algorithms estimate the label of each node based on a

smoothness assumption on the graph. Specifically, the algorithm estimates a class attachment "score" $f_i$ for each node $i$ by solving the optimization problem:

$$\min_{\mathbf{f}} \quad \omega_{ii'} \left( f_i - f_{i'} \right)^2 \tag{10}$$

$$s.t. \quad f_i = y_i \; \forall \; 1 \le i \le n_\ell. \tag{11}$$

Here, the term $\omega_{ii'} \left( f_i - f_{i'} \right)^2$ imposes label consistency of nearby samples (smoothness condition on the labels of the graph). The optimization problem in Equation (10) is the classical Laplacian regularization algorithm studied in depth in (Mai and Couillet, 2018). There, the authors showed the fundamental importance to "center" the weight matrix $\mathbf{W}$. This centering approach corrects an important bias in the regularized Laplacian which completely annihilates the use of unlabeled data in a large dimensional setting. A significant performance increase was reported, both in theory and in practice in (Mai and Couillet, 2021) when this basic, yet counter-intuitive, correction is accounted for. More specifically the centering is performed as follows

$$\hat{\mathbf{W}} = \mathbf{P}\mathbf{W}\mathbf{P}$$

with $\mathbf{P} = \left( \mathbf{I}_n - \frac{1}{n} \mathbb{1}_n \mathbb{1}_n^\top \right)$ the centering projector.

However, the optimization problem described in (10) now becomes non convex since the entries of the weight matrix $\mathbf{W}$ may take negative values (this must actually be the case as the mean value of the entries of $\mathbf{W}$ is zero). To deal with this problem, Mai and Couillet (2021) proposes to constrain the norm of the unlabeled data score vector $\mathbf{f}_u$ (that is, the score vector restricted to unlabeled data) by appending a regularization term $\alpha \|\mathbf{f}_u\|^2$ to the previous minimization problem. This leads, under a more convenient matrix formulation, to

$$\min_{\mathbf{f}_u \in \mathbb{R}^{n_u}} \alpha \|\mathbf{f}_u\|^2 - \mathbf{f}^\top \hat{\mathbf{W}} \mathbf{f} \qquad s.t. \; \mathbf{f}_\ell = \mathbf{y}_\ell. \tag{12}$$

This problem is now convex for all $\alpha > \|\hat{\mathbf{W}}_{uu}\|$ where $\hat{\mathbf{W}}_{uu}$ is the restriction of the matrix $\hat{\mathbf{W}}$ to the unlabeled data.

The optimization problem is a quadratic optimization problem with linear equality constraints, and $\mathbf{f}_u$ can be obtained explicitly and its solutions are best described under the matrix formulation (using a linear kernel, *i.e.,* $h(x) = x$):

$$\mathbf{f}_u = \frac{1}{n} \mathbf{y}_\ell^\top \mathbf{X}_\ell^\top \left( \lambda \mathbf{I}_d - \frac{1}{n} \mathbf{X}_u \mathbf{X}_u^\top \right)^{-1} \mathbf{X}_u. \tag{13}$$

The graph-based SSL solution given in Equation (13) is a particular case of QLDS solution given in Equation (9) with $\alpha_u = 1$ and $\alpha_\ell = 0$.

## C.2 Link to Spectral Clustering and choice of $\lambda$

Spectral clustering is a particular case of (13) when $\lambda$ is the maximum eigenvalue of $\frac{1}{n} \mathbf{X}_u \mathbf{X}_u^\top$. Indeed, using the eigenvalue decomposition $\frac{1}{n} \mathbf{X}_u \mathbf{X}_u^\top = \mathbf{U}\mathbf{\Lambda}\mathbf{U}^\top = \sum_{i=1}^{d} \lambda_i \mathbf{u}_i \mathbf{u}_i^\top$, Equation (13) can be rewritten as

$$\mathbf{f}_u = \frac{1}{n} \mathbf{y}_\ell^\top \mathbf{X}_\ell^\top \left( \lambda \mathbf{I}_d - \frac{1}{n} \mathbf{X}_u \mathbf{X}_u^\top \right)^{-1} \mathbf{X}_u$$

$$= \frac{1}{n} \sum_{i=1}^{d} \mathbf{y}_\ell \mathbf{X}_\ell^\top \frac{\mathbf{u}_i \mathbf{u}_i^\top}{\lambda - \lambda_i} \mathbf{X}_u.$$

When $\lambda \to \lambda_{max} = \max \lambda_i$ and denoting by $\mathbf{u}_{max}$ the eigenvector corresponding to the largest eigenvalue $\lambda_{max}$, we obtain,

$$\mathbf{f}_u \sim \mathbf{y}_\ell \mathbf{X}_\ell^\top \frac{\mathbf{u}_{max} \mathbf{u}_{max}^\top}{\lambda - \lambda_{max}} \mathbf{X}_u$$

$$\propto \mathbf{u}_{max}^\top \mathbf{X}_u,$$

which unfolds as projecting[1] the unlabeled data $\mathbf{X}_u$ into the largest eigenvector of $\frac{1}{n} \mathbf{X}_u \mathbf{X}_u^\top$ corresponding to the spectral clustering algorithm with linear kernel.

---

[1] up to a scaling $\mathbf{y}_\ell^\top \mathbf{X}_\ell^\top \mathbf{u}_{max}$ which does not impact the classification error

**The parameter** $\lambda$    In order for QLDS to specialize to spectral clustering in the unlabelled regime, we fix the parameter $\lambda = \lambda_{\max}$ to be the maximum eigenvalue of $\mathbf{X} = [\mathbf{X}_\ell, \mathbf{X}_u]$. For numerical reasons, in all the experiments we use $\lambda = (1 + \varepsilon)\lambda_{\max}$ with $\varepsilon = 10^{-3}$. Although many choices of $\varepsilon$ have been tried out, we do not find substantial improvements at considering it as an hyper-parameter and therefore fix it.

## D    THEORETICAL ANALYSIS OF QLDS

We recall the solution of the optimization problem of QLDS as

$$\mathbf{f}_u = \frac{1}{n}\mathbf{y}_\ell^\top \mathbf{X}_\ell^\top \left( \lambda \mathbf{I}_d + \alpha_\ell \frac{\mathbf{X}_\ell \mathbf{X}_\ell^\top}{n} - \alpha_u \frac{\mathbf{X}_u \mathbf{X}_u^\top}{n} \right)^{-1} \mathbf{X}_u. \tag{14}$$

The goal is to understand the statistical behavior of $\mathbf{f}_u$ in particular its distribution, and the moments of the distribution. To that end, we will assume the following concentration property on the data $\mathbf{X} = [\mathbf{X}_\ell, \mathbf{X}_u]$.

**Assumption 3 (Distribution of $\mathcal{D}(\mathbf{X})$)** *There exist two constants $C, c > 0$ (independent of $n, d$) such that, for any 1-Lipschitz function $f : \mathbb{R}^{p \times n} \to \mathbb{R}$,*

$$\mathbb{P}_{\mathbf{x} \sim \mathcal{D}(\mathbf{X})} \left( |f(\mathbf{x}) - m_{f(\mathbf{x})}| \geq t \right) \leq Ce^{-(t/c)^2} \quad \forall t > 0,$$

*where $m_Z$ is a median of the random variable $Z$. We require that the columns of $\mathbf{X}$ are independent and that for $\ell \in \{1, 2\}$, $\mathbf{x}_1^{(\ell)}, \ldots, \mathbf{x}_{n_\ell}^{(\ell)}$ are i.i.d. such that $\mathrm{Cov}(\mathbf{x}_i^{(\ell)}) = \mathbf{\Sigma}_\ell$. We further denote the mean and covariance for the columns of $\mathbf{X}$ respectively as $\boldsymbol{\mu}_\ell \equiv \mathbb{E}[\mathbf{x}_1^{(\ell)}]$ and $\mathbf{C}_\ell = \mathbf{\Sigma}_\ell + \boldsymbol{\mu}_\ell \boldsymbol{\mu}_\ell^\top$.*

As discussed in the main article, Assumption 3 notably encompasses the following scenarios: the columns of $\mathbf{X}$ are (i) independent Gaussian random vectors with identity covariance, (ii) independent random vectors uniformly distributed on the $\mathbb{R}^p$ sphere of radius $\sqrt{p}$, and, most importantly, (iii) any Lipschitz continuous transformation thereof. Scenario (iii) is of particular relevance for practical data settings as it was recently shown (Seddik et al., 2020). Indeed, random data generated by GANs (for example, images) can be modeled as in case (iii).

An intuitive explanation of Assumption 1 is that the transformed random variables $f(\mathbf{x})$ for any $f : \mathbb{R}^d \to \mathbb{R}$ Lipschitz has a variance of order $\mathcal{O}(1)$. In particular, it implies that it does not depend on the initial dimension $d$. Although we are not aware of any formal method to check whether some data follow this assumption, a line of reasoning suggests that this concentration property is most likely present in many real data. Indeed, most machine learning algorithms are Lipschitz applications that transform data of high dimension $d$ into a scalar (the decision score). If the data were not concentrated the decision score $f(x)$ would have a very large variance (depending on the dimension $d$) which would in turn lead to a random performance. The fact that a machine algorithm is supposed to obtain non-trivial performance (different from randomness) combined with the fact that common machine learning algorithms are Lipschitz applications suggests that the concentration assumption is not meaningless for real applications.

As an example, we perform the following experiment: for the `books` data set, we take a subset of examples and a subset of features, learn `QLDS(1,0)` on them, and plot the empirical distribution of $f(\mathbf{x})$. With conduct this experiment with the increasing $n$ and $d$, and see in Figure 3 that the variance with this increase remains to be of the same order.

Furthermore, we place ourselves into the following large dimensional regime.

**Assumption 4 (Growth Rate)** *As $n \to \infty$, we consider the regime where $d = \mathcal{O}(n)$, we assume $d/n \to c_0 > 0$. Furthermore, $n_{\ell j}/n \to c_{\ell j}$ and $n_{uj}/n \to c_{uj}$ for $j = 1, 2$. We denote by $\mathbf{c}_\ell = [c_{\ell 1}, c_{\ell 2}]$ and $\mathbf{c}_u = [c_{u1}, c_{u2}]$.*

This assumption of the commensurable relationship between the number of samples and their dimension corresponds to a realistic regime and differs from classical asymptotic where the number of samples is often assumed to be exponentially larger than the feature size, which is very unlikely in real-life applications.

Under Assumptions 3 and 4, the objective of this section is three-folds: (i) determine the distribution of $\mathbf{f}_u$ (ii) determine the first order moment of $\mathbf{f}_u$ and (iii) determine the second order moments of $\mathbf{f}_u$.

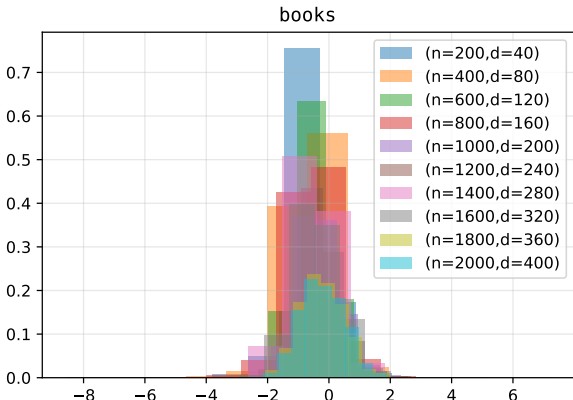

Figure 3: Practical illustration of the concentration property on `books` data set. With increasing $n$ and $d$, every colored histogram corresponds to the empirical distribution of the $g(\mathbf{x})$ of `QLDS(1,0)`.

### D.1  DISTRIBUTION OF $\mathbf{f}_u$

The proof of the gaussian distribution of the decision score of several learning schemes has been provided recently in (Tiomoko et al., 2020) (for the theoretical analysis of Multi-Task Learning), (Seddik et al., 2021) (in the case of the theoretical analysis of softmax). We follow a similar approach that is described as follows.

**Proof under Gaussian mixture model.**    Under a Gaussian mixture assumption for the input data $\mathbf{X}$, the convergence in distribution of the statistics of the classification score $f(\mathbf{x})$ is immediate as the projection of the deterministic vector $\boldsymbol{\omega}$ on the Gaussian random vector $\mathbf{x}$, it follows that $\boldsymbol{\omega}^\top \mathbf{x}$ is asymptotically Gaussian.

**Extension to concentrated random vector assumption.**    Since conditionally on the training data $\mathbf{X}$, the classification score $g(\mathbf{x})$ is expressed as the projection of the deterministic vector $\boldsymbol{\omega}$ on the concentrated random vector $\mathbf{x}$, the CLT for concentrated vector unfolds by proving that projections of deterministic vector on concentrated random vector is asymptotically gaussian. This is ensured by the following result.

**Theorem 3 (CLT for concentrated vector (Klartag, 2007; Fleury et al., 2007))** *If $\mathbf{x}$ is a concentrated random vector as defined in Assumption 1 with $\mathbb{E}[\mathbf{x}] = 0$, $\mathbb{E}[\mathbf{x}\mathbf{x}^\top] = I_p$ and $\sigma$ be the uniform measure on the sphere $\mathcal{S}^{p-1} \subset \mathbb{R}^p$ of radius 1, then for any integer $k = \mathcal{O}(1)$, there exist two constants $C, c$ and a set $\Theta \subset (\mathcal{S}^{p-1})^k$ such that $\underbrace{\sigma \otimes \ldots \otimes \sigma}_{k}(\Theta) \geq 1 - \sqrt{p}Ce^{-c\sqrt{p}}$ and*

$\forall \theta = (\theta_1, \ldots, \theta_k) \in \Theta$,

$$\forall a \in \mathbb{R}^k : \sup_{t \in \mathbb{R}} |\mathbb{P}(a^\top \theta^\top \mathbf{x} \geq t) - G(t)| \leq Cp^{-\frac{1}{4}},$$

*with $G(t)$ the cumulative distribution function of $\mathcal{N}(0, 1)$.*

Then the result unfolds naturally. Since $g(\mathbf{x})$ is asymptotically Gaussian, we will focus on computing its first and second order moment.

### D.2  FIRST ORDER MOMENT OF $\mathbf{f}_u$

Using Equation (14), the first order moment of $\mathbf{f}_u$ can be computed as

$$\mathbb{E}[\mathbf{f}_u] = \mathbb{E}\left[\frac{1}{n}\mathbf{y}_\ell^\top \mathbf{X}_\ell^\top \left(\lambda \mathbf{I}_d + \alpha_\ell \frac{\mathbf{X}_\ell \mathbf{X}_\ell^\top}{n} - \alpha_u \frac{\mathbf{X}_u \mathbf{X}_u^\top}{n}\right)^{-1} \mathbf{X}_u\right]. \tag{15}$$

Let's define for convenience the data matrix $\mathbf{X}$ being the concatenation of the labeled and unlabeled data matrix $\mathbf{X}_\ell$ and $\mathbf{X}_u$, *i.e.*, $\mathbf{X} = [\mathbf{X}_\ell, \mathbf{X}_u] \in \mathbb{R}^{d \times n}$. Then the expectation in (15) can be rewritten in the more convenient compact formulation

$$\mathbb{E}[\mathbf{f}_u] = \mathbb{E}[\frac{1}{n}\mathbf{y}_\ell^\top \mathbf{X}_\ell^\top \mathbf{Q}\mathbf{X}_u], \quad \mathbf{Q} = \left(\lambda\mathbf{I}_d + \frac{\mathbf{X}\mathbf{A}\mathbf{X}^\top}{n}\right)^{-1}, \quad \mathbf{A} = \begin{pmatrix} \alpha_\ell\mathbf{I}_{n_\ell} & \mathbf{0}_{n_\ell \times n_u} \\ \mathbf{0}_{n_u \times n_\ell} & -\alpha_u\mathbf{I}_{n_u} \end{pmatrix}.$$

To proceed, we furthermore introduce the matrices $\mathbf{S}_\ell = \begin{pmatrix} \mathbf{I}_{n_\ell} \\ \mathbf{0}_{n_u \times n_\ell} \end{pmatrix} \in \mathbb{R}^{n \times n_\ell}$ and $\mathbf{S}_u = \begin{pmatrix} \mathbf{I}_{n_u} \\ \mathbf{0}_{n_\ell \times n_u} \end{pmatrix} \in \mathbb{R}^{n \times n_u}$ such that $\mathbf{X}_\ell = \mathbf{X}\mathbf{S}_\ell$, and $\mathbf{X}_u = \mathbf{X}\mathbf{S}_u$. This lead to the following compact expression depending only on the random matrix $\mathbf{X}$

$$\mathbb{E}[\mathbf{f}_u] = \mathbb{E}[\mathbf{y}_\ell^\top \mathbf{S}_\ell^\top \frac{\mathbf{X}^\top \mathbf{Q}\mathbf{X}}{n}\mathbf{S}_u]. \tag{16}$$

Furthermore, let us recall the concept of *deterministic equivalents*, a classical object in random matrix theory.

**Definition 2** ((Couillet and Debbah, 2011, Chapter 6)) *A deterministic matrix $\bar{\mathbf{F}} \in \mathbb{R}^{n \times d}$ is said to be a* deterministic equivalent *of a given random matrix $\mathbf{F} \in \mathbb{R}^{n \times d}$, denoted $\bar{\mathbf{F}} \leftrightarrow \mathbf{F}$, if for any deterministic linear functional $f_{n,p} : \mathbb{R}^{n \times d} \to \mathbb{R}$ of bounded norm (uniformly over $d, n$), $f_{n,p}(\mathbf{F} - \bar{\mathbf{F}}) \to 0$ almost surely as $n, d \to \infty$.*

In particular if $\bar{\mathbf{F}} \leftrightarrow \mathbf{F}$, then $\mathbf{u}^\top(\mathbf{F} - \bar{\mathbf{F}})\mathbf{v} \xrightarrow{\text{a.s.}} 0$ for $\mathbf{u}, \mathbf{v}$ two unit vectors, and for all deterministic matrix $\mathbf{A}$ of bounded norm we also have $\frac{1}{n} \operatorname{tr} \mathbf{A}(\mathbf{F} - \bar{\mathbf{F}}) \xrightarrow{\text{a.s.}} 0$.

Deriving deterministic equivalents of the various objects under consideration will be a crucial tool to derive the main result. In particular, deterministic equivalents are particularly suitable to handle bilinear forms involving the random matrix $\mathbf{F}$, in particular for the statistics of $\mathbf{f}_u$ where the bilinear form $\frac{\mathbf{X}^\top \mathbf{Q}\mathbf{X}}{n}$ appears (see Equation (16)).

**Deterministic equivalent of $\frac{\mathbf{X}^\top \mathbf{Q}\mathbf{X}}{n}$** Let $\mathbf{u}, \mathbf{v}$ unit vectors for the $\ell_2$-norm, we develop:

$$\frac{1}{n}\mathbb{E}[\mathbf{u}^\top \mathbf{X}^\top \mathbf{Q}\mathbf{X}\mathbf{v}] = \frac{1}{n}\sum_{i,j=1}^n \mathbb{E}\left[u_i\mathbf{x}_i^\top \mathbf{Q}\mathbf{x}_j v_j\right]$$

$$= \frac{1}{n}\sum_{i=1}^n \mathbb{E}\left[u_i\mathbf{x}_i^\top \mathbf{Q}\mathbf{x}_i v_i\right] + \frac{1}{n}\sum_{\substack{i,j=1 \\ i \neq j}}^n \mathbb{E}\left[u_i\mathbf{x}_i^\top \mathbf{Q}\mathbf{x}_j v_j\right].$$

Furthermore, let us define for convenience the matrix $\mathbf{X}_{-i}$, which is the matrix $\mathbf{X}$ with a vector of $\mathbf{0}_p$ on its $i$-th column such that $\mathbf{X}\mathbf{X}^\top = \mathbf{X}_{-i}\mathbf{X}_{-i}^\top + \mathbf{x}_i\mathbf{x}_i^\top$. Applying the Sherman-Morrison matrix inversion lemma (*i.e.*, , $(\mathbf{M} + \mathbf{u}\mathbf{v}^\top)^{-1} = \mathbf{M}^{-1} - \frac{\mathbf{M}^{-1}\mathbf{u}\mathbf{v}^\top\mathbf{M}^{-1}}{1 + \mathbf{v}^\top\mathbf{M}^{-1}\mathbf{u}}$ for any invertible matrix $\mathbf{M}$ and vectors $\mathbf{u}, \mathbf{v}$) to $\mathbf{Q}$ leads to

$$\mathbf{Q} = \mathbf{Q}_{-i} - \frac{1}{n}\frac{A_{ii}\mathbf{Q}_{-i}\mathbf{x}_i\mathbf{x}_i^\top \mathbf{Q}_{-i}}{1 + \frac{1}{n}A_{ii}\mathbf{x}_i^\top \mathbf{Q}_{-i}\mathbf{x}_i}, \quad \mathbf{Q}_{-i} = \left(\frac{\mathbf{X}_{-i}\mathbf{A}\mathbf{X}_{-i}^\top}{n} + \lambda\mathbf{I}_d\right)^{-1}.$$

The latter allows to disentangle the strong dependency between $\mathbf{Q}$ and $\mathbf{x}_i$ as

$$\mathbf{Q}\mathbf{x}_i = \frac{\mathbf{Q}_{-i}\mathbf{x}_i}{1 + \frac{1}{n}A_{ii}\mathbf{x}_i^\top \mathbf{Q}_{-i}\mathbf{x}_i}. \tag{17}$$

Using Equation (17) we rewrite $\frac{\mathbf{X}^\top \mathbf{Q}\mathbf{X}}{n}$ as

$$\frac{1}{n}\mathbb{E}[\mathbf{u}^\top \mathbf{X}^\top \mathbf{Q}\mathbf{X}\mathbf{v}] = \frac{1}{n}\sum_{i=1}^n \mathbb{E}\left[\frac{u_i\mathbf{x}_i^\top \mathbf{Q}_{-i}\mathbf{x}_i v_i}{1 + A_{ii}\bar{\delta}_i}\right] + \frac{1}{n}\sum_{\substack{i,j=1 \\ i \neq j}}^n \mathbb{E}\left[\frac{u_i\mathbf{x}_i^\top \mathbf{Q}_{-ij}\mathbf{x}_j v_j}{(1 + A_{ii}\bar{\delta}_i)(1 + A_{jj}\bar{\delta}_j)}\right],$$

with $\bar{\delta}_i = \frac{1}{n}\mathbb{E}\left[\mathbf{x}_i^\top \mathbf{Q}_{-i}\mathbf{x}_i\right]$. Assumption 3 ensures that $\mathbf{x}_1^{(j)}, \ldots, \mathbf{x}_{n_k}^{(j)}$, $j = 1, 2$, are i.i.d. data vectors, we impose the natural constraint of equal $\bar{\delta}_1 = \ldots = \bar{\delta}_{n_k}$ within every class $j = 1, 2$. As such, we may reduce the complete score vector $\bar{\boldsymbol{\delta}} \in \mathbb{R}^n$ under the form

$$\bar{\boldsymbol{\delta}} = [\delta_1 \mathbb{1}_{n_{\ell 1}}^\top, \delta_2 \mathbb{1}_{n_{\ell 2}}^\top, \delta_1 \mathbb{1}_{n_{u 1}}^\top, \delta_2 \mathbb{1}_{n_{u 2}}^\top]^\top, \tag{18}$$

where $\delta_j = \frac{1}{n}\mathbb{E}\left[\mathbf{x}_i^\top \mathbf{Q}_{-i}\mathbf{x}_i | \mathbf{x}_i \in \mathcal{C}_j\right] = \frac{1}{n}\operatorname{tr}(\boldsymbol{\Sigma}_j \bar{\mathbf{Q}})$ is defined for each class $j = 1, 2$.

Using the shortcut notation $\bar{\mathbf{x}}_i = \mathbb{E}[\mathbf{x}_i]$ and the independence between samples $\mathbf{x}_i$ and $\mathbf{x}_j$ for $i \neq j$, the expectation can finally be obtained as

$$\frac{1}{n}\mathbb{E}[\mathbf{u}^\top \mathbf{X}^\top \mathbf{Q}\mathbf{X}\mathbf{v}] = \sum_{i=1}^n \frac{u_i \bar{\delta}_i v_i}{1 + A_{ii}\bar{\delta}_i} + \frac{1}{n}\sum_{\substack{i,j=1 \\ i\neq j}}^n \frac{u_i \bar{\mathbf{x}}_i^\top \bar{\mathbf{Q}}_{-ij} \bar{\mathbf{x}}_j v_j}{(1 + A_{ii}\bar{\delta}_i)(1 + A_{jj}\bar{\delta}_j)} + \mathcal{O}(1/\sqrt{n}).$$

We therefore deduce a deterministic equivalent for $\mathbf{X}^\top \mathbf{Q}\mathbf{X}$ as

$$\frac{1}{n}\mathbf{X}^\top \mathbf{Q}\mathbf{X} \leftrightarrow \boldsymbol{\Delta} + \frac{1}{n}\mathbf{J}\mathbf{M}_\delta^\top \bar{\mathbf{Q}}\mathbf{M}_\delta \mathbf{J}^\top,$$

where $\boldsymbol{\Delta}$ is the diagonal matrix $\Delta_{ii} = \frac{\bar{\delta}_i}{1 + A_{ii}\bar{\delta}_i}$, $\mathbf{M}_\delta = [\frac{\boldsymbol{\mu}_1}{1 + \alpha_\ell \delta_1}, \frac{\boldsymbol{\mu}_2}{1 + \alpha_\ell \delta_2}, \frac{\boldsymbol{\mu}_1}{1 - \alpha_u \delta_1}, \frac{\boldsymbol{\mu}_2}{1 - \alpha_u \delta_2}]$ and

$$\mathbf{J} = \begin{pmatrix} \mathbb{1}_{n_{\ell 1}} & & 0 \\ & \ddots & \\ 0 & & \mathbb{1}_{n_{u 2}} \end{pmatrix}.$$

The expectation can finally be obtained as

$$\mathbb{E}[\mathbf{f}_u] = \mathbf{y}_\ell^\top \mathbf{S}_\ell^\top \left(\boldsymbol{\Delta} + \frac{1}{n}\mathbf{J}\mathbf{M}_\delta^\top \bar{\mathbf{Q}}\mathbf{M}_\delta \mathbf{J}^\top\right)\mathbf{S}_u$$

$$= \frac{1}{n}\mathbf{y}_\ell^\top \mathbf{S}_\ell^\top \mathbf{J}\mathbf{M}_\delta^\top \bar{\mathbf{Q}}\mathbf{M}_\delta \mathbf{J}^\top \mathbf{S}_u.$$

It then remains to find a deterministic equivalent $\bar{\mathbf{Q}}$ for $\mathbf{Q}$. Similarly as performed in (Louart and Couillet, 2018), the deterministic equivalent for $\mathbf{Q}$ can be obtained as

$$\mathbf{Q} \leftrightarrow \bar{\mathbf{Q}} = \left(\lambda \mathbf{I}_d + \frac{\alpha_\ell c_{\ell 1} \mathbf{C}_1}{1 + \alpha_\ell \delta_1} + \frac{\alpha_\ell c_{\ell 2} \mathbf{C}_2}{1 + \alpha_\ell \delta_2} - \frac{\alpha_u c_{u 1} \mathbf{C}_1}{1 - \alpha_u \delta_1} - \frac{\alpha_u c_{u 2} \mathbf{C}_2}{1 - \alpha_u \delta_2}\right)^{-1}. \tag{19}$$

Further defining $\kappa_1 = \frac{c_{\ell 1}\alpha_\ell}{1 + \alpha_\ell \delta_1} - \frac{c_{u 1}\alpha_u}{1 - \alpha_u \delta_1}$, $\kappa_2 = \frac{c_{\ell 2}\alpha_\ell}{1 + \alpha_\ell \delta_2} - \frac{c_{u 2}\alpha_u}{1 - \alpha_u \delta_2}$, we can further write

$$\bar{\mathbf{Q}} = \bar{\mathbf{Q}}_0 - \bar{\mathbf{Q}}_0 \mathbf{M}^\top (\mathcal{D}_\kappa^{-1} + \mathbf{M}^\top \bar{\mathbf{Q}}_0 \mathbf{M})\mathbf{M}^\top \bar{\mathbf{Q}}_0, \quad \bar{\mathbf{Q}}_0 = (\lambda \mathbf{I}_d + \kappa_1 \boldsymbol{\Sigma}_1 + \kappa_2 \boldsymbol{\Sigma}_2)^{-1}.$$

Therefore $\mathbf{M}_\delta^\top \bar{\mathbf{Q}}\mathbf{M}_\delta = \mathcal{D}_{\tilde{\delta}}\mathcal{A}^\top \mathcal{D}_\kappa^{-1}\left[\mathbf{I}_2 - \left(\mathcal{D}_\kappa^{-1} + \mathbf{M}^\top \bar{\mathbf{Q}}_0 \mathbf{M}\right)^{-1}\mathcal{D}_\kappa^{-1}\right]\mathcal{A}\mathcal{D}_{\tilde{\delta}}$. where $\tilde{\boldsymbol{\delta}} = [1/(1 + \alpha_\ell \delta_1), 1/(1 + \alpha_\ell \delta_2), 1/(1 - \alpha_u \delta_1), 1/(1 - \alpha_u \delta_2)]$ and $\mathcal{A} = [\mathbf{I}_2, \mathbf{I}_2]$.

We then deduce the expectation as

$$m_j = \mathbb{E}[f_i | \mathbf{x}_i \in \mathcal{C}_j] = (\mathbf{e}_1^{[2]} - \mathbf{e}_2^{[2]})^\top \mathcal{D}_{\mathbf{c}_\ell}\mathcal{D}_{\tilde{\boldsymbol{\delta}}_\ell}\mathcal{D}_\kappa^{-1}\left[\mathbf{I}_2 - \left(\mathcal{D}_\kappa^{-1} + \mathbf{M}^\top \bar{\mathbf{Q}}_0 \mathbf{M}\right)^{-1}\mathcal{D}_\kappa^{-1}\right]\mathcal{D}_{\tilde{\boldsymbol{\delta}}_u}\mathbf{e}_j^{[2]}$$

with $\mathcal{M} = \left(\mathcal{D}_\kappa^{-1} + \mathbf{M}^\top \bar{\mathbf{Q}}_0 \mathbf{M}\right)^{-1}$, $\tilde{\boldsymbol{\delta}}_\ell = [1/(1 + \alpha_\ell \delta_1), 1/(1 + \alpha_\ell \delta_2)]$ and $\tilde{\boldsymbol{\delta}}_u = [1/(1 - \alpha_u \delta_1), 1/(1 - \alpha_u \delta_2)]$.

In the case of identity covariance tackled in the main article we have $\delta := \delta_1 = \delta_2$ and

$$\mathcal{M} = \left(\mathcal{D}_\kappa^{-1} + \frac{\mathbf{M}^\top \mathbf{M}}{\lambda + \kappa_1 + \kappa_2}\right)^{-1}. \tag{20}$$

Therefore the mean reads as

$$\boxed{m_\ell = \mathbb{E}[f_i | \mathbf{x}_i \in \mathcal{C}_j] = \frac{(-1)^j \left(c_{\ell j} - (\mathbf{e}_1^{[2]} - \mathbf{e}_2^{[2]})^\top \mathcal{D}_{\mathbf{c}_\ell}\mathcal{D}_\kappa^{-1}\mathcal{M}\mathbf{e}_j^{[2]}\right)}{\kappa_j (1 - \alpha_u \delta)(1 + \alpha_\ell \delta)}.} \tag{21}$$

### D.3 Second order moment of $\mathbf{f}_u$

The second order moment of $\mathbf{f}_u$ can be computed as

$$\mathbb{E}[\mathbf{f}_u^\top \mathbf{f}_u] = \frac{1}{n^2}\mathbb{E}\left[\mathbf{y}_\ell^\top \mathbf{S}_\ell^\top \mathbf{X}^\top \mathbf{Q}\mathbf{X}\mathbf{S}_u\mathbf{S}_u^\top \mathbf{X}^\top \mathbf{Q}\mathbf{X}\mathbf{S}_\ell \mathbf{y}_\ell\right].$$

Let's define by convenience the matrix $\mathbf{B} = \mathbf{S}_u\mathbf{S}_u^\top$. As previously we are looking for a deterministic equivalent for $\mathbf{X}^\top \mathbf{Q}\mathbf{X}\mathbf{B}\mathbf{X}^\top \mathbf{Q}\mathbf{X}$. We proceed in the same way by computing $\frac{1}{n^2}\mathbb{E}[\mathbf{u}^\top \mathbf{X}^\top \mathbf{Q}\mathbf{X}\mathbf{B}\mathbf{X}^\top \mathbf{Q}\mathbf{X}\mathbf{v}]$ for all $\mathbf{u}, \mathbf{v}$ of unit norm:

$$\frac{1}{n^2}\mathbb{E}[\mathbf{u}^\top \mathbf{X}^\top \mathbf{Q}\mathbf{X}\mathbf{B}\mathbf{X}^\top \mathbf{Q}\mathbf{X}\mathbf{v}] = \frac{1}{n^2}\sum_{i,j,k=1}^{n} u_i\mathbf{x}_i^\top \mathbf{Q}\mathbf{x}_j B_{jj}\mathbf{x}_j^\top \mathbf{Q}\mathbf{x}_k v_k$$

$$= \frac{1}{n^2}\sum_{i=1}^{n}\mathbb{E}[u_i\mathbf{x}_i^\top \mathbf{Q}\mathbf{x}_i B_{ii}\mathbf{x}_i^\top \mathbf{Q}\mathbf{x}_i v_i] + \frac{1}{n^2}\sum_{\substack{i,j,k=1\\i\neq j\neq k}}^{n}\mathbb{E}[u_i\mathbf{x}_i^\top \mathbf{Q}\mathbf{x}_j B_{jj}\mathbf{x}_j^\top \mathbf{Q}\mathbf{x}_k v_k]$$

$$+ \frac{1}{n^2}\sum_{\substack{i,k=1\\i\neq k}}^{n}\mathbb{E}[u_i\mathbf{x}_i^\top \mathbf{Q}\mathbf{x}_i B_{ii}\mathbf{x}_i^\top \mathbf{Q}\mathbf{x}_k v_k] + \frac{1}{n^2}\sum_{\substack{i,j=1\\i\neq j}}^{n}\mathbb{E}[u_i\mathbf{x}_i^\top \mathbf{Q}\mathbf{x}_j B_{jj}\mathbf{x}_j^\top \mathbf{Q}\mathbf{x}_j v_j]$$

$$+ \frac{1}{n^2}\sum_{\substack{i,j=1\\i\neq j}}^{n}\mathbb{E}[u_i\mathbf{x}_i^\top \mathbf{Q}\mathbf{x}_j B_{jj}\mathbf{x}_j^\top \mathbf{Q}\mathbf{x}_i v_i],$$

and we reuse Equation (17) in order to continue

$$= \frac{1}{n^2}\sum_{i}^{n}\frac{\mathbb{E}[u_i\mathbf{x}_i^\top \mathbf{Q}_{-i}\mathbf{x}_i B_{ii}\mathbf{x}_i^\top \mathbf{Q}_{-i}\mathbf{x}_i v_i]}{(1+A_{ii}\bar{\delta}_i)^2} + \frac{1}{n^2}\sum_{i\neq j\neq k}^{n}\frac{\mathbb{E}[u_i\mathbf{x}_i^\top \mathbf{Q}_{-ij}\mathbf{x}_j B_{jj}\mathbf{x}_j^\top \mathbf{Q}_{-jk}\mathbf{x}_k v_k]}{(1+A_{ii}\bar{\delta}_i)(1+A_{jj}\bar{\delta}_j)^2(1+A_{kk}\bar{\delta}_k)}$$

$$+ \frac{2}{n^2}\sum_{i\neq j}^{n}\frac{\mathbb{E}[u_i\mathbf{x}_i^\top \mathbf{Q}_{-ij}\mathbf{x}_j B_{jj}\mathbf{x}_j^\top \mathbf{Q}_{-j}\mathbf{x}_j v_j]}{(1+A_{ii}\bar{\delta}_i)(1+A_{jj}\bar{\delta}_j)^2} + \frac{1}{n^2}\sum_{i\neq j}^{n}\frac{\mathbb{E}[u_i\mathbf{x}_i^\top \mathbf{Q}_{-ij}\mathbf{x}_j B_{jj}\mathbf{x}_j^\top \mathbf{Q}_{-ij}\mathbf{x}_i v_i]}{(1+A_{ii}\bar{\delta}_i)^2(1+A_{jj}\bar{\delta}_j)^2} + \mathcal{O}(1/\sqrt{n})$$

$$= \sum_{i}^{n} u_i\frac{\bar{\delta}_i^2}{(1+A_{ii}\bar{\delta}_i)^2}B_{ii}v_i + \frac{1}{n}\sum_{i\neq k}^{n}\frac{u_i\bar{\mathbf{x}}_i^\top \mathbf{Q}\mathbf{C}_{u\delta}\mathbf{Q}\bar{\mathbf{x}}_k v_k}{(1+A_{ii}\bar{\delta}_i)(1+A_{kk}\bar{\delta}_k)}$$

$$+ \frac{2}{n}\sum_{i\neq j}^{n}\frac{u_i\bar{\mathbf{x}}_i^\top \bar{\mathbf{Q}}\bar{\mathbf{x}}_j\bar{\delta}_j B_{jj}v_j}{(1+A_{ii}\bar{\delta}_i)(1+A_{jj}\bar{\delta}_j)^2} + \frac{1}{n}\sum_{i=1}^{n}\frac{\operatorname{tr}(\mathbf{C}_i\mathbf{Q}\mathbf{C}_{u\delta}\mathbf{Q})u_i v_i}{(1+A_{ii}\delta_i)^2} + \mathcal{O}(1/\sqrt{n}),$$

where $\mathbf{C}_{u\delta}$ is defined as

$$\mathbf{C}_{u\delta} = \frac{1}{n}\sum_{i=1}^{n}\frac{B_{ii}\mathbf{x}_i\mathbf{x}_i^\top}{(1+A_{ii}\bar{\delta}_i)^2} = \frac{c_{uj}\mathbf{C}_j}{(1-\alpha_u\delta_j)^2}. \tag{22}$$

Let's denote for convenience by $\mathbf{E}$ the deterministic equivalent of $\mathbf{Q}\mathbf{C}_{u\delta}\mathbf{Q}$, then a deterministic equivalent for $\mathbf{X}^\top \mathbf{Q}\mathbf{X}\mathbf{B}\mathbf{X}^\top \mathbf{Q}\mathbf{X}$ is given as :

$$\frac{1}{n^2}\mathbf{X}^\top \mathbf{Q}\mathbf{X}\mathbf{B}\mathbf{X}^\top \mathbf{Q}\mathbf{X} \leftrightarrow \mathbf{\Delta}^2\mathbf{B} + \frac{\mathbf{J}\mathbf{M}_\delta^\top \mathbf{E}\mathbf{M}_\delta\mathbf{J}^\top}{n} + 2\frac{\mathbf{\Delta}\mathbf{B}\mathbf{J}\mathbf{M}_\delta^\top \bar{\mathbf{Q}}\mathbf{M}_\delta\mathbf{J}^\top}{n} + \mathcal{E}$$

where $\mathcal{E}$ is the diagonal matrix containing on its diagonal $\mathcal{E}_{ii} = \frac{1}{n}\mathbb{E}[\operatorname{tr}(\mathbf{C}_i\mathbf{Q}\mathbf{C}_{u\delta}\mathbf{Q})] = \frac{1}{n}\operatorname{tr}(\mathbf{C}_i\mathbf{E})$.

We therefore deduce the variance of $\mathbf{f}_u$ as

$$\operatorname{Var}(\mathbf{f}_u) = \mathbb{E}[\mathbf{f}_u^\top \mathbf{f}_u] - \mathbb{E}[\mathbf{f}_u]^2$$

$$= \mathbf{y}_\ell^\top \mathbf{S}_\ell^\top \left(\mathbf{\Delta}^2\mathbf{B} + \frac{\mathbf{J}\mathbf{M}_\delta^\top \mathbf{E}\mathbf{M}_\delta\mathbf{J}^\top}{n} + 2\frac{\mathbf{\Delta}\mathbf{B}\mathbf{J}\mathbf{M}_\delta^\top \bar{\mathbf{Q}}\mathbf{M}_\delta\mathbf{J}^\top}{n} + \mathcal{E}\right)\mathbf{S}_\ell\mathbf{y}_\ell$$

$$- \mathbf{y}_\ell^\top \mathbf{S}_\ell^\top \left(\mathbf{\Delta} + \frac{1}{n}\mathbf{J}\mathbf{M}_\delta^\top \bar{\mathbf{Q}}\mathbf{M}_\delta\mathbf{J}^\top\right)\mathbf{B}\left(\mathbf{\Delta} + \frac{1}{n}\mathbf{J}\mathbf{M}_\delta^\top \bar{\mathbf{Q}}\mathbf{M}_\delta\mathbf{J}^\top\right)S_\ell\mathbf{y}_\ell$$

$$= \mathbf{y}_\ell^\top \mathbf{S}_\ell^\top \left(\frac{\mathbf{J}\mathbf{M}_\delta^\top \mathbf{E}\mathbf{M}_\delta\mathbf{J}^\top}{n} + \mathcal{E}\right)\mathbf{S}_\ell\mathbf{y}_\ell.$$

The last step consists in finding a deterministic equivalent of $\mathbf{Q}\mathbf{C}_{u\delta}\mathbf{Q}$ denoted $\mathbf{E}$. To that end let's evaluate for any deterministic vector $\mathbf{v}, \mathbf{u} \in \mathbb{R}^d$ of unit norm, $\frac{1}{n}\mathbb{E}[\mathbf{u}^\top \mathbf{Q}\mathbf{C}_{u\delta}(\mathbf{Q} - \bar{\mathbf{Q}})\mathbf{v}]$. Applying the matrix identity $\mathbf{A}^{-1} - \mathbf{B}^{-1} = \mathbf{A}^{-1}(\mathbf{B} - \mathbf{A})\mathbf{B}^{-1}$ for any invertible matrix $\mathbf{A}, \mathbf{B}$ to $\mathbf{Q} - \bar{\mathbf{Q}}$ and using algebraic simplifications in particular Equation (17) allow to successively obtain

$$\frac{1}{n}\mathbb{E}\left[\mathbf{u}^\top \mathbf{Q}\mathbf{C}_{u\delta}(\mathbf{Q} - \bar{\mathbf{Q}})\mathbf{v}\right] = \frac{1}{n}\mathbb{E}\left[\mathbf{u}^\top \mathbf{Q}\mathbf{C}_{u\delta}\mathbf{Q}\left(\mathbf{C}_\delta - \frac{\mathbf{X}\mathbf{A}\mathbf{X}^\top}{n}\right)\bar{\mathbf{Q}}\mathbf{v}\right]$$

$$= \frac{1}{n}\sum_i^n \mathbb{E}\left[-\frac{1}{n}\frac{A_{ii}\mathbf{u}^\top \mathbf{Q}\mathbf{C}_{u\delta}\mathbf{Q}_{-i}\mathbf{x}_i\mathbf{x}_i^\top \bar{\mathbf{Q}}\mathbf{v}}{1 + A_{ii}\bar{\delta}_i} + \mathbf{u}^\top \mathbf{Q}\mathbf{C}_{u\delta}\mathbf{Q}\mathbf{C}_\delta \bar{\mathbf{Q}}\mathbf{v}\right] + \mathcal{O}(1/\sqrt{n})$$

$$= \frac{1}{n}\sum_i^n \mathbb{E}\left[-\frac{1}{n}\frac{A_{ii}\mathbf{u}^\top \mathbf{Q}_{-i}\mathbf{C}_{u\delta}\mathbf{Q}_{-i}\mathbf{x}_i\mathbf{x}_i^\top \bar{\mathbf{Q}}\mathbf{v}}{1 + A_{ii}\bar{\delta}_i}\right] + \frac{1}{n^2}\sum_i^n \mathbb{E}\left[\frac{1}{n}\frac{A_{ii}^2\mathbf{u}^\top \mathbf{Q}_{-i}\mathbf{x}_i\mathbf{x}_i^\top \mathbf{Q}_{-i}\mathbf{C}_{u\delta}\mathbf{Q}_{-i}\mathbf{x}_i\mathbf{x}_i^\top \bar{\mathbf{Q}}\mathbf{v}}{(1 + A_{ii}\bar{\delta}_i)^2}\right] + \mathcal{O}(1/\sqrt{n})$$

$$= \frac{1}{n^2}\sum_i^n \mathbb{E}\left[\frac{1}{n}\operatorname{tr}\left(\mathbf{C}_i\mathbf{Q}_{-i}\mathbf{C}_{u\delta}\mathbf{Q}\right)\frac{A_{ii}^2\mathbf{u}^\top \bar{\mathbf{Q}}\mathbf{C}_i\bar{\mathbf{Q}}\mathbf{v}}{(1 + A_{ii}\bar{\delta}_i)^2}\right] + \mathcal{O}(1/\sqrt{n}),$$

where $\bar{\mathbf{Q}} = (\lambda\mathbf{I}_d + \mathbf{C}_\delta)^{-1}$, *i.e.*, $\mathbf{C}_\delta = \frac{\alpha_\ell c_{\ell 1}\mathbf{C}_1}{1 + \alpha_\ell \delta_1} + \frac{\alpha_\ell c_{\ell 2}\mathbf{C}_2}{1 + \alpha_\ell \delta_2} - \frac{\alpha_u c_{u1}\mathbf{C}_1}{1 - \alpha_u \delta_1} - \frac{\alpha_u c_{u2}\mathbf{C}_2}{1 - \alpha_u \delta_2}$.

Therefore

$$\mathbf{Q}\mathbf{C}_{u\delta}\mathbf{Q} \leftrightarrow \mathbf{E} = \bar{\mathbf{Q}}\mathbf{C}_{u\delta}\bar{\mathbf{Q}} + \sum_{k=1}^2 \frac{c_{\ell k}\alpha_\ell^2 d_k}{(1 + \alpha_\ell \delta_k)^2}\bar{\mathbf{Q}}\mathbf{C}_k\bar{\mathbf{Q}} + \sum_{k=1}^2 \frac{c_{uk}\alpha_u^2 d_k}{(1 - \alpha_u \delta_k)^2}\bar{\mathbf{Q}}\mathbf{C}_k\bar{\mathbf{Q}} \qquad (23)$$

where $d_k = \frac{1}{n}\operatorname{tr}(\mathbf{C}_k\mathbf{Q}\mathbf{C}_{u\delta}\mathbf{Q})$.

Right Multiplying Equation (23) by $\mathbf{C}_k$ and taking the trace allows to retrieve an expression for $\mathbf{D} = \mathcal{D}_\mathbf{d}$, with $\mathbf{d} = [d_1, d_2]$ as

$$\mathbf{D} = \mathcal{D}_{\bar{\mathbf{t}}}\left(\mathbf{I}_2 - \mathcal{D}_{\tilde{\mathbf{a}}}\tilde{\mathcal{V}}\right)^{-1}, \quad \tilde{\mathcal{V}}_{kk'} = \frac{1}{n}\operatorname{tr}\left(\mathbf{C}_k\bar{\mathbf{Q}}\mathbf{C}_{k'}\bar{\mathbf{Q}}\right),$$

$$\bar{t}_k = \frac{1}{n}\operatorname{tr}\left(\mathbf{C}_k\bar{\mathbf{Q}}\mathbf{C}_{u\delta}\bar{\mathbf{Q}}\right), \quad \tilde{a}_k = \frac{c_{\ell k}\alpha_\ell^2}{(1 + \alpha_\ell \delta_k)^2} + \frac{c_{uk}\alpha_u^2}{(1 - \alpha_u \delta_k)^2}.$$

Similarly as performed for the mean, the variance can be furthermore simplified as

$$\operatorname{Var}(f_i) = (\mathbf{e}_1 - \mathbf{e}_2)^\top \left(\mathcal{D}_{\mathbf{c}_\ell}\mathcal{D}_{\tilde{\delta}}\mathcal{D}_\kappa^{-1}\mathcal{M}\mathcal{G}\mathcal{M}\mathcal{D}_\kappa^{-1}\mathcal{D}_{\tilde{\delta}}\mathcal{D}_{\mathbf{c}_\ell} + \mathcal{D}_\mathbf{d}\mathcal{D}_{\mathbf{c}_\ell}\right)(\mathbf{e}_1 - \mathbf{e}_2)$$

where $\mathcal{G} = \mathbf{M}^\top \bar{\mathbf{Q}}_0 \bar{\mathbf{C}}\bar{\mathbf{Q}}_0 \mathbf{M}$, $\bar{\mathbf{C}} = \mathbf{C}_{u\delta} + \sum_{k=1}^2 \tilde{a}_k d_k \mathbf{C}_k$. In the case of identity covariance matrix tackled in the main article,

$$\mathcal{G} = \left(-\frac{c_u}{(1 - \alpha_u \delta)} + \sum_{k=1}^2 \tilde{a}_k d_k\right)\delta\mathbf{M}^\top \mathbf{M}$$

with $d_k$ and $\tilde{a}_k$ which simplifies as

$$d_k = -\frac{1}{(1 - \alpha_u \delta)^2}\left(\frac{c_0 c_u}{(\lambda + \kappa_1 + \kappa_2)^2 - c_0 \tilde{a}_k}\right), \quad \tilde{a}_k = \frac{c_{\ell k}\alpha_\ell^2}{(1 + \alpha_\ell \delta)^2} + \frac{c_{uk}\alpha_u^2}{(1 - \alpha_u \delta)^2}.$$

This leads to the theorem in the general covariance matrix

**Theorem 4** *Let $\mathbf{X} \in \mathbb{R}^{d \times n}$ be a data set that follows Assumptions 3 and 4 and consider the notation convention defined previously. For any $\mathbf{x} \in \mathbf{X}_u$ with $\mathbf{x} \in \mathcal{C}_j$ and $f(\mathbf{x}) = \frac{1}{\sqrt{n}}\boldsymbol{\omega}^{\star\top}\mathbf{x}$, we have almost surely for both classes $j$*

$$f(\mathbf{x}|\mathbf{x} \in \mathcal{C}_j) - \mathfrak{f}_j \xrightarrow{\text{a.s.}} 0, \quad \text{where} \quad \mathfrak{f}_j \sim \mathcal{N}\left(m_j, {\sigma_j}^2\right).$$

*The mean $m_j$ and the variance $\sigma^2$ are defined as*

$$m_j = \frac{1}{n}\mathbf{y}_\ell^\top \mathbf{S}_\ell^\top \mathbf{J}\mathbf{M}_\delta^\top \bar{\mathbf{Q}}\mathbf{M}_\delta \mathbf{J}^\top \mathbf{S}_u,$$

$$\sigma_j^2 = (\mathbf{e}_1 - \mathbf{e}_2)^\top \left(\mathcal{D}_{\mathbf{c}_\ell}\mathcal{D}_{\tilde{\delta}}\mathcal{D}_\kappa^{-1}\mathcal{M}\mathcal{G}\mathcal{M}\mathcal{D}_\kappa^{-1}\mathcal{D}_{\tilde{\delta}}\mathcal{D}_{\mathbf{c}_\ell} + \mathcal{D}_\mathbf{d}\mathcal{D}_{\mathbf{c}_\ell}\right)(\mathbf{e}_1 - \mathbf{e}_2),$$

# E EXPERIMENTS

This section complements Section 5 of the main paper by giving more details of the experimental setup and performing two additional experiments.

## E.1 EXPERIMENTAL SETUP

Table 3 sums up the characteristics of publicly available real data sets used in our experiments. As we are interested in the practical use of the proposed approach in the semi-supervised regime, we test the performance in the case when $n_l \ll n_u$. Thus, instead of using the original train/test splits proposed by data sources, we set our own labeled/unlabeled splits to fit the semi-supervised context. For each data set, we perform an experiment 20 times by randomly splitting original data on a labeled and an unlabeled sets fixing their sample sizes to the values shown in Table 3. For the results, we evaluate the transductive error on the unlabeled data and display the average and the standard deviation (both in %) over the 20 trials. All experiments were performed on a laptop with an Intel(R) Core(TM) i7-8565U CPU @ 1.80GHz, 16GB RAM. The implementation code for reproducing the experimental results of the paper will be released upon acceptance of the article.

Table 3: Characteristics of data sets used in our experiments.

| Data set | # of lab. examples, | # of unlab. examples, | Dimension, | Class Proportions |
|---|---|---|---|---|
| Books | 20 | 1980 | 400 | 0.5:0.5 |
| DVD | 19 | 1980 | 400 | 0.5:0.5 |
| Electronics | 19 | 1979 | 400 | 0.5:0.5 |
| Kitchen | 19 | 1980 | 400 | 0.5:0.5 |
| Splice | 10 | 990 | 60 | 0.48:0.52 |
| Mushrooms | 81 | 8043 | 112 | 0.48:0.52 |
| Adult | 325 | 32236 | 14 | 0.76:0.24 |

## E.2 ESTIMATION OF CLASS PROPORTIONS FOR UNLABELED DATA

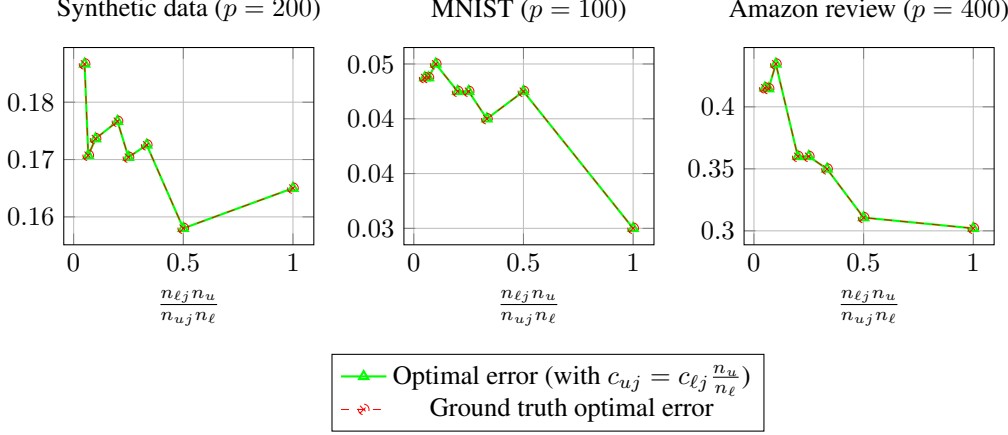

Figure 4: Optimal Classification error as a function of discrepancy between class proportion in labeled and unlabeled set ($\frac{n_{\ell j} n_u}{n_{uj} n_\ell}$).

In the first experiment, we analyze the influence of the assumption $c_{uj} = c_{\ell j} \frac{n_u}{n_\ell}$ (proportion of class 1 and class 2 is the same in unlabeled and labeled set) by representing the optimal classification under this assumption and the optimal classification error knowing the true value of $c_{uj}$ as function of the violation of this assumption represented by the ratio $\frac{n_{\ell j} n_u}{n_{uj} n_\ell}$. As a recall, this assumption is needed since the theoretical performance depends on $c_{uj}$ which is not known a priori and needs to

be estimated. As shown in Figure 4, this assumption doesn't alter the overall behavior of the model selection approach since the model selection is the same even though the proportion of class 1 and 2 are different in labeled set and unlabeled set ($\frac{n_{\ell j}n_u}{n_{uj}n_\ell} \neq 1$)

### E.3 IMPROVEMENT OVER THE SUPERVISED BASELINE

In a second experiment, we represent the gain with respect to LSSVM ($\varepsilon(\alpha_l^\star, \alpha_u^\star) - \varepsilon(\alpha_l^\star, \alpha_u^\star)$) of QLDS when optimizing the hyperparameters (as performed theoretically in Algorithm 1 of the main paper) as function of the difficulty of the task (implemented through the norm of the matrix $\mathcal{M}$) and the number of labeled samples. Figure 5 which looks like a "phase diagram" shows that a non-trivial gain is obtained with respect to a fully supervised case. In particular, one can see that the gain of using a semi-supervised approach is relevant when few labeled samples are available and when the task is difficult. This conclusion is similar to existing conclusion from (Mai and Couillet, 2021; Lelarge and Miolane, 2019).

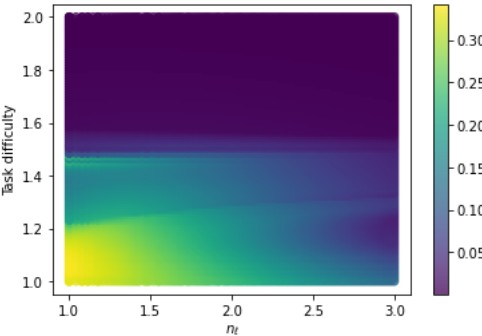

Figure 5: (**Left**) Relative gain with respect to supervised learning as a function of the labeled sample size and the task difficulty (through the choice of the distance between the mean of class 1 and class 2 $\|\mu_1 - \mu_2\|$) on synthetic gaussian mixture model. A higher value of $\|\mu_1 - \mu_2\|$ means that the task is easy and a smaller value means that the task is difficult. On the left lower corner (difficult task and a small number of labeled samples ) a non-trivial gain is obtained with respect to fully supervised case. The task difficulty in the $y$-axis is $\|\mu_1 - \mu_2\|$ which measures the distance between the mean of the two classes.

### E.4 COMPARISON OF DIFFERENT SEMI-SUPERVISED LOSSES

In this section, we additionally support our choice of the learning objective given by Eq. 1 and compare different possibilities to construct the loss function for semi-supervised linear classification. More specifically, we compare for the labeled part

1. the quadratic loss $\sum_{i=1}^{n_\ell}(y_i - \mathbf{x}_i^\top \boldsymbol{\omega})^2$,
2. differentiable surrogate of the hinge loss $\sum_{i=1}^{n_\ell} \frac{1}{\gamma}\log(1 + \exp\{\gamma(1 - y_i\mathbf{x}_i^\top \boldsymbol{\omega})\})$ with $\gamma$ set to 20 (Zhang and Oles, 2001),
3. the log-loss $\sum_{i=1}^{n_\ell} y_i \log \sigma(\mathbf{x}_i^\top \boldsymbol{\omega}) + (1 - y_i)\log(1 - \sigma(\mathbf{x}_i^\top \boldsymbol{\omega}))$,

and for the unlabeled part

1. the quadratic margin $\sum_{i=n_\ell+1}^{n_\ell+n_u}(\boldsymbol{\omega}^\top \mathbf{x}_i)^2$,
2. the differentiable surrogate of the absolute value of the margin $\sum_{i=n_\ell+1}^{n_\ell+n_u} \exp\{-3(\boldsymbol{\omega}^\top \mathbf{x}_i)^2\}$ (Chapelle and Zien, 2005).

We consider all possible combinations of the labeled and the unlabeled parts which result in 6 semi-supervised losses. We optimize them using Adam optimizer (Kingma and Ba, 2015) fixing the learning rate and the weight decay to $10^{-3}$ and $10^{-5}$, respectively. Note that when the square loss and the quadratic margin are considered, we have a gradient-based version of QLDS.

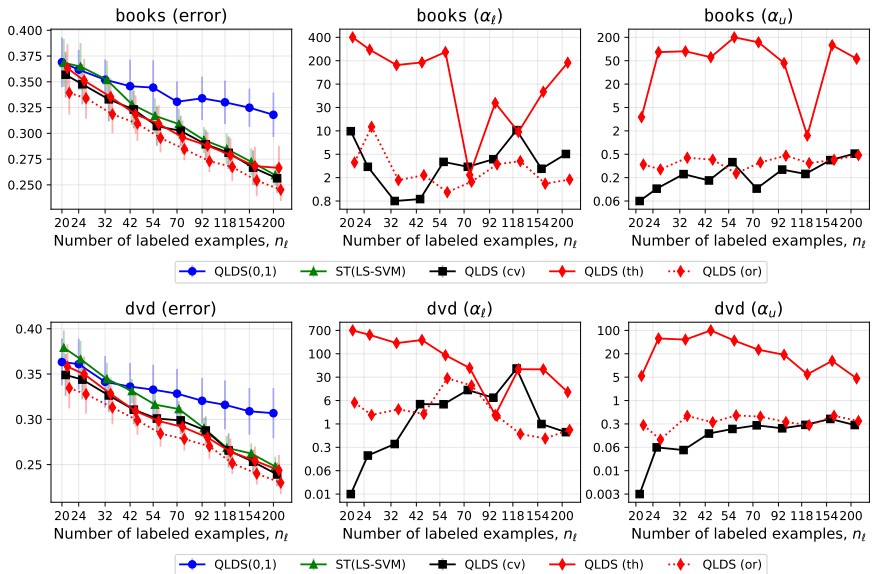

Figure 6: The performance and model selection results on different data sets with the increase of the number of labeled examples.

For fair comparison, for each loss, we perform a grid search over possible values of $\alpha_\ell, \alpha_u, \lambda$ and choose the best solution according to the oracle, namely, the performance on the unlabeled data. Table 4 illustrates the performance results on 7 real data sets. One can see that in most of cases the quadratic margin outperforms the absolute value of the margin. In general, the combination of the square loss and the quadratic margin appears to be stable leading to the second-best solution in many cases. Thus, by choosing this learning objective, we do not lose much efficiency, having a convex objective and the ability to conduct theoretical analysis.

Table 4: The classification error of different semi-supervised losses on the real benchmark data sets. Square Loss - Quadratic Margin corresponds to QLDS. The smallest and the second smallest error values are highlighted in bold and italics, respectively.

| Data set | Square Loss | | Hinge Loss | | Log-Loss | |
|---|---|---|---|---|---|---|
| | Quadratic | Abs Value | Quadratic | Abs Value | Quadratic | Abs Value |
| books | $25.82 \pm 1.23$ | $34.13 \pm 2.71$ | $\mathbf{23.83} \pm 0.85$ | $33.38 \pm 2.78$ | $36.2 \pm 1.99$ | $36.67 \pm 2.05$ |
| dvd | $24.81 \pm 2.97$ | $34.74 \pm 2.76$ | $\mathbf{23.33} \pm 1.9$ | $34.86 \pm 2.72$ | $37.34 \pm 2.37$ | $37.69 \pm 2.22$ |
| electronics | $19.82 \pm 0.57$ | $26.07 \pm 2.88$ | $\mathbf{19.22} \pm 0.56$ | $26.37 \pm 2.89$ | $32.52 \pm 2.55$ | $33.27 \pm 2.83$ |
| kitchen | $18.75 \pm 0.85$ | $24.02 \pm 2.86$ | $\mathbf{17.93} \pm 0.57$ | $22.95 \pm 1.84$ | $31.04 \pm 3.04$ | $31.75 \pm 3.3$ |
| splice | $34.47 \pm 2.59$ | $\mathbf{34.29} \pm 3.8$ | $34.42 \pm 2.23$ | $34.3 \pm 3.73$ | $38.7 \pm 2.26$ | $38.72 \pm 2.27$ |
| mushrooms | $1.55 \pm 0.9$ | $\mathbf{1.17} \pm 0.66$ | $2.33 \pm 1.02$ | $1.75 \pm 0.74$ | $1.9 \pm 0.86$ | $1.98 \pm 1.0$ |
| adult | $19.63 \pm 0.88$ | $19.65 \pm 0.9$ | $\mathbf{18.38} \pm 0.73$ | $18.5 \pm 0.81$ | $18.43 \pm 0.64$ | $18.47 \pm 0.72$ |

### E.5 PERFORMANCE DEPENDING ON THE NUMBER OF LABELED EXAMPLES

This section extends Section 5.2 of the main paper by providing experimental results for different size of labeled set. In addition, we depict the values of $\alpha_l$ and $\alpha_u$ (averaged over 20 splits) taken by QLDS (th), QLDS (cv) and QLDS (or). All the results can be seen in Figure 6 and Figure 7.

### E.6 CHOICE OF $\lambda$

In Section 5, we have fixed the value of $\lambda$ as the maximum eigenvalue of $\mathbf{X} = [\mathbf{X}_\ell, \mathbf{X}_u]$ for all versions of QLDS. To support this choice and make sure that it does not harm the baselines, in this section we provide an additional experiment, where we compare the fixed value of $\lambda$ with the case

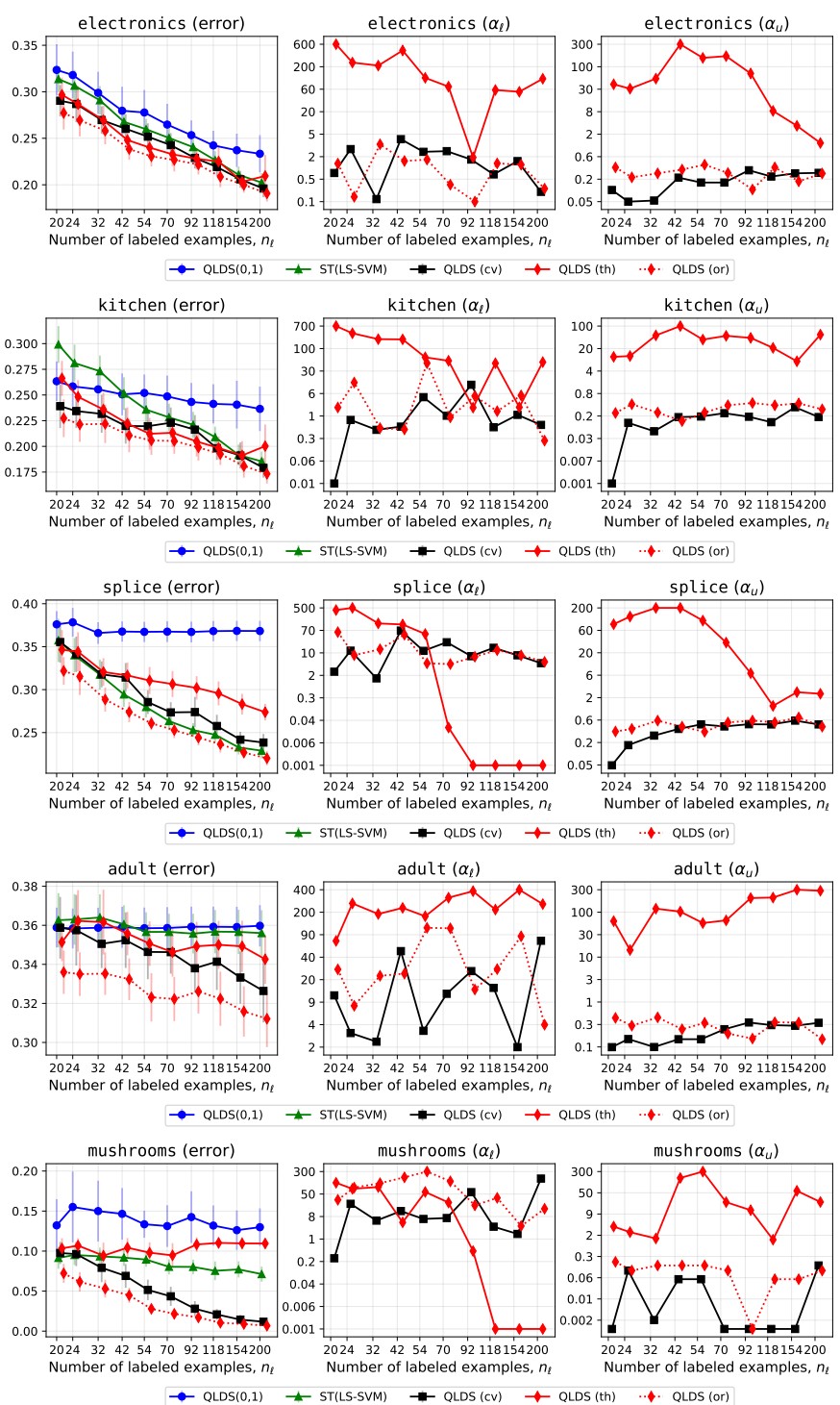

Figure 7: The performance and model selection results on different data sets with the increase of the number of labeled examples.

when $\lambda$ is tuned by the 10-fold cross-validation on the available labeled set. Table 5 depicts this comparison for QLDS(1,0) (LS-SVM) and QLDS(0,1) (GB-SSL). As one can see on 5 of 7 data sets the maximum eigenvalue heuristics outperforms the cross-validation. The experimental results suggests that the cross-validation policy is more relevant for the cases where the labeled data is more informative than unlabeled data (adult and mushrooms). Otherwise, the maximum eigenvalue heuristics seems to be more appropriate, which is accorded with (Mai and Couillet, 2021).

Table 5: The classification error of the supervised and the unsupervised baselines when the hyperparameter $\lambda$ is fixed to the maximum eigenvalue, and when it's tuned using the cross-validation on the labeled set. The smallest error for each baseline is highlighted in bold.

| Data set | QLDS(1,0) (LS-SVM) | | QLDS(0,1) (GB-SSL) | |
|---|---|---|---|---|
| | Fixed | CV | Fixed | CV |
| books | $\mathbf{37.47} \pm 2.25$ | $38.32 \pm 2.37$ | $\mathbf{26.47} \pm 0.72$ | $32.84 \pm 8.65$ |
| dvd | $\mathbf{38.33} \pm 1.72$ | $38.56 \pm 2.03$ | $\mathbf{29.12} \pm 1.35$ | $32.74 \pm 7.26$ |
| electronics | $\mathbf{34.15} \pm 3.25$ | $35.2 \pm 3.0$ | $\mathbf{19.4} \pm 0.29$ | $23.8 \pm 9.19$ |
| kitchen | $\mathbf{32.39} \pm 3.02$ | $33.42 \pm 4.44$ | $\mathbf{19.31} \pm 0.16$ | $22.05 \pm 8.55$ |
| splice | $\mathbf{39.81} \pm 2.93$ | $40.38 \pm 3.31$ | $\mathbf{35.48} \pm 0.86$ | $39.53 \pm 3.53$ |
| adult | $33.35 \pm 0.68$ | $\mathbf{32.13} \pm 1.88$ | $36.28 \pm 0.06$ | $\mathbf{34.0} \pm 0.73$ |
| mushrooms | $6.55 \pm 2.07$ | $\mathbf{2.53} \pm 1.38$ | $11.33 \pm 0.04$ | $\mathbf{8.8} \pm 1.47$ |

