# OpenReview forum: "Random Matrix Analysis to Balance between Supervised and Unsupervised Learning under the Low Density Separation Assumption"
_ICLR.cc/2023/Conference — Submitted to ICLR 2023_

### Official Review · Reviewer_Cq3e · 2022-10-22

**Confidence:** 3
**Correctness:** 4
**Technical Novelty And Significance:** 3
**Empirical Novelty And Significance:** 3
**Recommendation:** 8

**Clarity, Quality, Novelty And Reproducibility:**

The proposed method bridges supervised and unsupervised learning - the least-square support vector machine in the supervised case, the spectral clustering in the fully unsupervised regime.

**Strength And Weaknesses:**

* Strength

The proposed method bridges supervised and unsupervised learning.

* Weakness

There is a lack of non-asymptotic results.

**Summary Of The Paper:**

The paper proposes a linear semi-supervised classification model, where the low density separation assumption is implemented via quadratic margin maximization.

It bridges supervised and unsupervised learning - the least-square support vector machine in the supervised case, the spectral clustering in the fully unsupervised regime.

**Summary Of The Review:**

The proposed method bridges supervised and unsupervised learning, and this is very new to me.

---

> ### Author Response · Authors · 2022-11-09
> **Answer to reviewer Cq3e**
>
> Thank you very much for the interest of our ideas.
>
> We agree with the reviewer that the analysis is performed under an asymptotic regime where $p,n\to\infty$ with $p/n<\infty$, which is classical asymptotic considered in Random Matrix Theory to define asymptotic statistical quantities. Although this growth regime is evaluated in an asymptotic setting, it is shown to be applicable for real and finite situations (real data) at a convergence rate of $\mathcal{O}(1/\sqrt{pn})$. In comparison, the classical asymptotics in statistics (suggested by the Central Theorem Limit as an example) which considers that $n\rightarrow\infty$ has convergence rates of the order of $\mathcal{O}(1/\sqrt{n})$.
> More intuitively, in Random Matrix Theory, by considering both randomness sample-wise and feature-wise, the concentration of statistical quantities of interest is faster ($\mathcal{O}(1/\sqrt{pn})$) than considering randomness sample-wise only (which leads to a rate of $\mathcal{O}(1/\sqrt{n})$ ). We would also like to mention that although the study is asymptotic, it provides exact results. To the best of our knowledge, a non-asymptotic study would derive a bound rather than an exact result. In our case, it would be possible to derive such bound with the convergence speed of $\mathcal{O}(1/\sqrt{pn})$, however, it will not lead to any further insight.

---

> > ### Author Response · Authors · 2022-11-25
> > **Message to reviewer Cq3e**
> >
> > Please accept our sincere thanks again for all your suggestions for our work. We hope our responses have answered your questions.
> > Do you have other questions? We will be happy to answer them.

---

### Official Review · Reviewer_xxab · 2022-10-23

**Confidence:** 4
**Correctness:** 3
**Technical Novelty And Significance:** 3
**Empirical Novelty And Significance:** 3
**Recommendation:** 6

**Clarity, Quality, Novelty And Reproducibility:**

The manuscript is clear and well-written, and to my best knowledge this is the second work in the literature providing sharp asymptotics for a semi-supervised setting (with [Lelarge, Miolane '19], who analysed the Bayes-optimal performance being the first).

The code to reproduce the plots is not provided with the manuscript.

**Details Of Ethics Concerns:**

N/A.

**Strength And Weaknesses:**

**Disclaimer**: For full transparency, I have previously reviewed this submission at a different venue where it was evaluated by four referees and the AC as a borderline rejection, with a recommendation to review and resubmit. Quoting the decision:

> *"Ultimately this was a borderline decision, and I reviewed the paper myself to make a final decision. I believe that the paper is in need of a major revision to address the issues raised by the reviewers, and thus cannot recommend acceptance at this time. But I do hope the authors revise and re-submit this paper to another top conference soon."*

Although the content of the work is essentially the same, I am happy to see that the authors have taken most of the suggestions raised by the reviewers into account in this new submission. In my (fresh) review below, I revisit some of the points previously discussed with the authors in view of further improving the submission.

**Strengths**: The paper is well-written and easy to follow. The exact asymptotic analysis of a semi-supervised task adds to a literature which has mostly focused on supervised and unsupervised settings.

**Weaknesses**: On the other hand, on a technical level this work doesn't bring any new significant technical contribution, but rather combine existing tools to analyse a task of interest.

**Comments**

- **[C1]**: I appreciate the authors have made an effort to diversify their bibliography. Just a small note on a new sentence in the *Related work* section:

>To continue with physical statistics-based methods, we highlight (Lelarge and Miolane, 2019) which derived Asymptotic Bayes
risk using theoretical information and a replica method.

[Lelarge, Miolane '19] *do not* employ the replica method. Instead, they do give a heuristic derivation of their result based on the *cavity method*, which is closely related to the "leave-one-out" method from RMT (and which I believe the authors are more familiar). This result is then proven using Guerra's interpolation technique [Guerra '03] and the Aizenman-Sims-Starr scheme [Aizenman et al. '03].

- **[C2]**: Although this is now mentioned en passant below eq. (1), the assumption that the regularisation $\lambda$ needs to be fixed to the top singular value of the features in order to make the empirical risk convex is very important for Theorem 1. This should be made more explicit in the assumptions.

- **[C3]**: In my previous review, I noted to the authors that the claim that it is a *"well-stablished fact that quadratic cost function are asymptotically optimal"* is both strong and inaccurate in the stated generality, since it is setting specific. For instance, for a simple binary classification task in a teacher-student setting with Gaussian covariates, optimally regularised ridge regression has a generalisation decay of $\sim n^{-1/2}$ while optimally regularised logistic regression has a generalisation decay rate of $\sim n^{-1}$, which matches the Bayes-optimal rate for this problem [Aubin et al. '20]. But other examples of settings where there is a clear benefit of using other losses over the square loss abound, e.g. random features model [Gerace et al. '20] or kernel classification with source and capacity conditions [Cui et al. '22] to mention a few.

The authors now mention they added a Table 4 in the appendix justifying this claim in their setting, but I was surprised to find Table 4 doesn't exist in the appendix.

Honestly, I don't understand why the authors insist in such a strong claim which is setting specific and wrong in general.


- **[C4]**: One of the key points of the paper is that SSL improves over the supervised baseline by using the theory motivated Algorithm 1 (QLDS). Therefore, a fair comparison should benchmark QLDS against both the fully supervised and unsupervised settings at *optimal* (and not fixed) regularisation (as done in the main text). Unless I missed it, this is only discussed in Appendix G:

>[...] Figure 5 which looks like a "phase diagram" shows that a non-trivial
gain is obtained with respect to a fully supervised case. In particular, one can see that the gain of
using a semi-supervised approach is relevant when few labeled samples are available and when the
task is difficult. This conclusion is similar to existing conclusion from (Mai and Couillet, 2021;
Lelarge and Miolane, 2019).

I believe the discussion of when one gains using SSL with respect to the baseline is quite a quite relevant outcome of this work. Therefore, I would strongly suggest the authors to mention it in the discussion of the theoretical results in the main text.


**Small typos**:

- *"[...] using theoretical information"* -> *"[...] using information theory"*

**References**:

[[Aizenman et al. '03]](https://journals.aps.org/prb/abstract/10.1103/PhysRevB.68.214403) M Aizenman, R Sims, and SL Starr. *Extended variational principle for the Sherrington-Kirkpatrick spin-glass model*. Physical Review B, 68(21):214403, 2003.

[[Guerra '03]](https://link.springer.com/article/10.1007/s00220-002-0773-5) F Guerra. *Broken replica symmetry bounds in the mean field spin glass model*. Communications
in mathematical physics, 233(1):1–12, 2003.

[[Aubin et al. '20]](https://proceedings.neurips.cc/paper/2020/hash/8f4576ad85410442a74ee3a7683757b3-Abstract.html) B Aubin, F Krzakala, Y Lu, L Zdeborová, "Generalization error in high-dimensional perceptrons: Approaching Bayes error with convex optimization", Part of Advances in Neural Information Processing Systems 33 (NeurIPS 2020)

[[Gerace et al. '20]](https://proceedings.mlr.press/v119/gerace20a.html) F Gerace, B Loureiro, F Krzakala, M Mézard, L Zdeborová, "Generalisation error in learning with random features and the hidden manifold model", Proceedings of the 37th International Conference on Machine Learning, PMLR 119:3452-3462, 2020.

[[Cui et al. '22]](https://arxiv.org/abs/2201.12655) H Cui, B Loureiro, F Krzakala, L Zdeborová, "Error Rates for Kernel Classification under Source and Capacity Conditions", arXiv: 2201.12655 [stat.ML]

**Summary Of The Paper:**

This work investigates the performance of a linear semi-supervised learning method named QLDS. The method in formulated in terms of an empirical risk minimisation problem consisting of the standard ridge risk (square loss + $\ell_2$ penalty) for the labelled data and a quadratic term for the unlabelled data encouraging low-density solutions. Since the risk is quadratic, one can write down an explicit solution, therefore mapping the question of computing the performance of QLDS to a random matrix theory problem. The key question of interest is how to optimally tune the hyperparameters gauging the trade-off between the supervised and unsupervised parts in order to minimise the population misclassification error. The main contributions are:

1. Under a concentration assumption on the data, to show that the predictor is an asymptotically Gaussian random variable with mean and standard deviation given by the solution of a set of self-consistent equations (Theorem 1). In particular, this can be used to derive an exact asymptotic formula for the misclassification error depending on the hyperparameters of the problem.

2. The equations above depend on the Gram matrix of the population means of the data. The second result provides a theoretical guarantee on the estimation of these means depending on the quantity of labelled data available (Proposition 2).

Together, these two results are used to study how to design a procedure to tune the hyperparameters in the model (relative strength between supervised and unsupervised terms), summarised in Algorithm 1. Some numerical experiments on both synthetic and real data are provided, comparing cross-validation to the procedure proposed, and the performance of optimally tuned QLDS to other methods from the literature.

**Summary Of The Review:**

This works provides a sharp asymptotic analysis of a semi-supervised setting, providing a theory-driven efficient algorithm to optimally tune the hyperparameters, and provides numerical support for the theory. Most of the concerns raised by the reviewers in the previous version have been fixed, and I hope the additional suggestions will further improve the work.

Overall, I believe this work is of interest to the ICLR community.

---

> ### Author Response · Authors · 2022-11-09
> **Answer to reviewer xxab**
>
> Thank you very much for your second review of our work. We indeed incorporated many modifications as suggested by the reviewers in the previous venue where we submitted this work and believe the paper greatly benefited from the discussions.
>
> **C1: On related works.**
> We thank the reviewer for this correct and very rewarding addition. We are indeed aware of the work on the leave-one-out approach widely used in random matrix theory and it would be very interesting as a future work to look for deep links between Lelarge's study and ours.
> We have corrected this sentence in the paper.
>
> **C2: On the relation between convexity and regularization parameter $\lambda$.**
> We agree with the reviewer that it is important to explicitly state the conditions for convexity of the loss.
> In view of the reviewer's comment, we have added a more explicit explanation in the appendix and added this clarification in the main text to the convexity statement.
> We provide more details about the convexity in our answer to reviewer LHdi.
>
> **C3: On the optimality of least square.**
> We totally agree with the reviewer that this statement has been shown in a very specific setting in random matrix theory. In the updated version, we give a more sober but correct statement. We believe it is a strong motivation for our choice of loss function.
> Thank you for pointing out the lack of Table 4 that we added (Appendix Section E.4), supporting the use of the quadratic losses.
>
> **C4: Choice of $\lambda$.** In the new version of the paper, we have added Section E.6, where we compare the performance when $\lambda$ is found as the maximum eigenvalue of data and when it is obtained via the cross-validation for both supervised and unsupervised baselines. The obtained experimental results suggest that the choice depends on the utility of the labeled data. While the cross-validation improves the performance of the baselines in some cases, the maximum eigenvalue heuristics have generally better results, which is accorded with **(Mai et al., 2021)**.
>
> **C4: On the usefulness of semi-supervised learning.**
> We agree with the reviewer that the intuition of usefulness of semi-supervised data in difficult regimes (small sample size and difficult task) deserves to be better highlighted, especially as it is widely shared by previous work.
> In the new version we have added this clarification as a complement to the explanation of the theorem. Although we are unable, due to space problems, to put the phase diagram figure in the appendix, we have referenced it.
>
> **Bibliography**
>  -  **(Mai et al., 2021)** Consistent Semi-Supervised Graph Regularization for High Dimensional Data. X. Mai, R. Couillet.
> J. Mach. Learn. Res. 22, 94:1-94:48.
>  -  **(Lelarge et al., 2019)** Asymptotic bayes risk for gaussian mixture in a semi-supervised setting. M. Lelarge, L. Miolane.
> 2019 IEEE 8th International Workshop on CAMSAP.

---

> > ### Author Response · Authors · 2022-11-25
> > **Message to reviewer xxab**
> >
> > Please accept our sincere thanks again for all your suggestions for our work. We hope our responses have answered your questions. Do you have other questions? We will be happy to answer them.

---

### Official Review · Reviewer_LHdi · 2022-10-31

**Confidence:** 3
**Correctness:** 2
**Technical Novelty And Significance:** 2
**Empirical Novelty And Significance:** 3
**Recommendation:** 3

**Clarity, Quality, Novelty And Reproducibility:**

Clarity: Okay. The mathematics in this paper are too sloppy for a paper whose main contribution is theoretical. A work like this should have a very polished and essentially self-contained theory. Some examples include:

1. (1) Despite the authors' claims, this is not convex (claim is made above (2)). This is seen easily by simply setting alpha_u to be large, in which case the risk is concave.
2. Theorem 3 in appendix. What is an "observable diameter"? This term appears nowhere else in the work and is quite mysterious.
3. Theorem 3 in appendix. What precisely is a "concentrated random vector"?
4. Theorem 3 in appendix. "small compared to p" is not precise.
5. Definition 2 in appendix. This limit isn't clear the function f is on a fixed space, but the dimension of the F matrices used as an input are changing. So does f also change? How do we reconcile this?

Quality: Again this is okay, however I don't find the theory to be of good quality.

Novelty: I am not aware of (1) existing elsewhere so it may be novel, however it is not much of a departure from existing formulations. The analysis of random matrix theory is straightforward and applying it in this sort of way has been done elsewhere (see citations in the paper).

Reproducibility: No code provided.

**Strength And Weaknesses:**

Strengths:
* Formulation is new as far as I know.
* Asymptotic analysis is nice.

Weaknesses:
* The theory isn't very well-explained and there are small mistakes or at least unclear statements in the math.
* The contributions aren't very significant: the experimental results don't include any strong competitors, the theory is a fairly straightforward application of existing results, and the classifier formulation isn't particularly novel.

**Summary Of The Paper:**

The authors of this paper present a semisupervised adaptation of the linear least squares-SVM. This formulation trivially contains the LS-SVM and linear spectral clustering as special cases. The bulk of the paper and perhaps its most interesting contribution is its theoretical analysis of the method which utilizes random matrix theory to analyze the asymptotic behavior of the estimator and gives a way to choose hyperparameters. The efficacy of the classifier is experimentally verified against some simple baselines.

**Summary Of The Review:**

It seems as though the topic presented here may be a valid line of research, but the paper presented here is not in state that I can recommend acceptance. Additionally the limited experimental evaluation and theoretical contributions are make this work very niche, I do not think this would be of significant interest to the ICLR crowd.

---

> ### Author Response · Authors · 2022-11-09
> **Answer to reviewer LHdi (Part 2)**
>
> **On the theoretical contributions.**
> Although the technical tools used in the present paper are already known (deterministic equivalent in particular), the goal and technical derivation are different from existing literature and require the computation of new expectation (in particular the mean and the variance).
> Furthermore, we would like to emphasize that no previous work in the literature attempted to understand theoretically Equation (2).
> The derivation of the mean, the variance and the different applications provided in paper is technically new and, in our opinion, sounded.
> The proofs provided in the supplementary material are quite technical and, in our opinion, far from being straightforward in the sense that our main theorem cannot be stated on purely intuitive arguments.
>
> **On the empirical contributions.**
> We would like to recall that our main contributions are: (i) an unification framework for studying the intrinsic and strong link between the algorithms usually analyzed independently (LSSVM, Spectral clustering, graph-based approaches) (ii) a subsequent theoretical analysis of the algorithm and (iii) an application of theory for automatically selecting the hyperparameters.
> From this point of view, we do not claim to create an algorithm that beats all the state-of-the-art but rather a proof of concept which justifies the choice of the baselines used as comparison.
> In our opinion, it would be totally unfair to compare ourselves with sophisticated approaches such as deep learning approaches that use a feature extraction step.
> However, in the updated version we provided the missing Table 4 where we experimentally compared our approach with 5 other semi-supervised linear classification losses that practically represent our main competitors. In addition, we extended our experimental part by comparing with the baselines for which $\lambda$ is tuned as suggested by reviewer xxab.

---

> ### Author Response · Authors · 2022-11-09
> **Answer to reviewer LHdi (part 1)**
>
> Thank you very much for the careful reading of our paper.
>
>
> **On the convexity.**
> We agree with the reviewer that the convexity holds under some conditions on the value of $\lambda$ that we did not emphasize enough in the paper, and it is explicitly stated in the updated version.
> Indeed one way to check for convexity is to compute the hessian of the loss given by $ \nabla\mathcal{L}({\mathbf \omega}) = \left(\lambda \mathbf{I}_d + \alpha_\ell\frac{\mathbf{X}_\ell\mathbf{X}_\ell^T}{n} - \alpha_u\frac{\mathbf{X}_u\mathbf{X}_u^T}{n} \right). $ Note that $\nabla\mathcal{L}({\mathbf\omega}) \geq 0$ if and only if $\lambda \geq \lambda_m\left(-\alpha_\ell\frac{\mathbf{X}_\ell\mathbf{X}_\ell^T}{n} + \alpha_u \frac{\mathbf{X}_u\mathbf{X}_u^T}{n}\right)$ where $\lambda_m(M)$ denotes the maximum eigenvalue of the matrix $M$.
> Therefore the loss function is convex as soon as  $\lambda \geq \lambda_m\left(-\alpha_\ell\frac{\mathbf{X}_\ell\mathbf{X}_\ell^T}{n} + \alpha_u \frac{\mathbf{X}_u\mathbf{X}_u^T}{n}\right)$. This can also be seen by looking at the optimal hyperplane of QLDS given as $\left(\lambda \mathbf{I}_d + \alpha_\ell\frac{\mathbf{X}_\ell\mathbf{X}_\ell^T}{n} - \alpha_u \frac{\mathbf{X}_u\mathbf{X}_u^T}{n}\right)^{-1}\frac{\mathbf{X}_\ell}{\sqrt{n}}\mathbf{y}_\ell$.
> In the new version of the paper (Appendix B), we have added more details with respect to this comment. In the main paper we have added this important precision as well.
>
>
> **Definition of some concepts.**
> *Concentrated random vectors* are defined at Assumption 1.
> We would like to point out that the concept of concentrated random vectors is a well studied object in the field of Random Matrix Theory and has been intensively discussed in related literature (see e.g. **(Talagrang et al., 1995})**, **(Louart et al., 2018)** for a complete literature on the subject).
> An intuitive explanation is that the transformed random variables $f(\mathbf x)$ for any $f: \mathbb{R}^{d} \rightarrow \mathbb{R}$ Lipschitz has a variance of order $\mathcal{O}(1)$.
> The variance of $f(\mathbf{x})$ is the so-called *observable diameter* rigorously defined in **(Chapter 3.1/2, Gromov et al., 1999)**. The observable diameter is the diameter of the “observations” of the distribution that could be seen as Lipschitz  projections of the concentrated random vector $\textbf{x}$.
> Imposing that the observable diameter is of order $\mathcal{O}(1)$ implies that it does not depend on the initial dimension $d$.
> For simplicity reason and thanks to the reviewer comment, we reformulate Theorem 3 by removing the notion of observable diameter as it is sufficient to refer to Assumption 1.
>
>
> **On additional details.**
> The term "small compared to" is used when performing asymptotic analysis (here in this case analyzing performance when $p\rightarrow\infty$). In asymptotic analysis, it is important to precise which quantity scales with $p$ (for example $n$ is assumed to scale linearly with $p$ while $k$ is supposed to be constant when $p\rightarrow\infty$). Therefore, $k$ is said to be "small compared to $p$" which means that $k$ doesn't scale linearly with $p$, i.e., $k=\mathcal{O}(1)$ when $p\rightarrow\infty$. In the updated version of the paper, we use the notation $k=\mathcal{O}(1)$ instead of the expression "small compare to ".
>
> In the definition 2 of appendix, the function $f$ is not defined on a fixed space (since $n,p\rightarrow\infty$). Therefore as correctly mentioned by the reviewer $f$ is a sequence of function and the almost sure convergence is understood for  $n,p\rightarrow\infty$.
> For more clarity, we will denote it as $f_{n,p}$.
>
> **Bibliography**
>
>  - **(Louart et al., 2018)** A random matrix approach to neural networks,
> C Louart, Z Liao, R Couillet
> The Annals of Applied Probability 28 (2), 1190-1248.
>  -  **(Gromov et al., 1999)** Metric structures for Riemannian and non-Riemannian spaces (Vol. 152). Boston: Birkhäuser. Gromov, M., Katz, M., Pansu, P., \& Semmes, S. (1999).
>  -  **(Talagrand et al., 1995)** Concentration of Measure and Isoperimetric
> Inequalities in product spaces. Publications mathématiques de
> l’I.H.E.S., tome 81, 1995. Michel Talagrand.
>  -  **(Seddik et al., 2020)** Random matrix theory proves that deep learning representations of GAN-data behave as gaussian mixtures
> MEA Seddik, C Louart, M Tamaazousti, R Couillet
> International Conference on Machine Learning, 8573-8582.

---

### Author Response · Authors · 2022-11-09
**Answer to all reviewers**

We would like to thank the reviewers for showing keen interest in our ideas, and for their thorough and very valuable comments.
We provide specific answers to each reviewer in response to their remarks. We also upload a new version of the paper and supplementary material incorporating in blue the updates.

---

### Decision · Program_Chairs · 2023-01-20

**Decision:**

Reject

**Justification For Why Not Higher Score:**

Concerns about proofs and overall lack of polish in the writing

**Justification For Why Not Lower Score:**

N/A

**Metareview: Summary, Strengths And Weaknesses:**

This paper presents a novel asymptotic analysis of semi-supervised learning based on ideas borrowed from random matrix theory. After multiple rounds of discussion, it was clear that this paper is not polished enough in its current form. There are too many typos and missing definitions, particularly in the proofs which at least one reviewer checked carefully. Reviewers agreed that another round of revision is needed.

In their discussion, the authors contend that some of these definitions are standard in other parts of the literature, but I can attest that many of these terms are not mainstream in ML. Since this submission is to a top ML conference, it is essential to properly motivate and define everything so that an ML researcher can verify the claims independently.

I urge the authors to carefully take the feedback from the reviewers into account and prepare a careful revision that defines everything carefully and corrects the many typos found by the reviewers.

**Summary Of Ac-Reviewer Meeting:**

Two points were discussed: 1) Concerns about writing/proofs, and 2) Suitability for ICLR. Consensus was reached that 1) is a serious concern, and that 2) is not a serious concern. Regarding 2), one reviewer pointed to several related papers in the ML literature on related ideas.